# On the Hyperparameter Loss Landscapes of Machine Learning Algorithms

## Abstract

Despite the recent success in a plethora of hyperparameter optimization (HPO) methods for machine learning (ML) models, the intricate interplay between model hyperparameters (HPs) and predictive losses (a.k.a *fitness*), which is a key prerequisite for understanding HPO, remain notably underexplored in our community. This results in limited explainability in the HPO process, rendering a lack of human trust and difficulties in pinpointing algorithm bottlenecks. In this paper, we aim to shed light on this black box by conducting large-scale *fitness landscape analysis* (FLA) on $1,500$ HP loss landscapes of $6$ ML models with more than $11M$ model configurations, across $67$ datasets and different levels of fidelities. We reveal the first unified, comprehensive portrait of their topographies in terms of smoothness, neutrality and modality. We also show that such properties are highly transferable across datasets and fidelities, providing fundamental evidence for the success of multi-fidelity and transfer learning methods. These findings are made possible by developing a dedicated FLA framework that incorporates a combination of visual and quantitative measures. We further demonstrate the potential of this framework by analyzing the NAS-Bench-101 landscape, and we believe it is able to faciliate fundamental understanding of a broader range of AutoML tasks.

## 1    Introduction

In the past decade, considerable efforts have been invested in developing hyperparameter optimization (HPO) techniques to automate the laborious task of hyperparameter (HP) tunning for machine learning (ML) models. Many successful approaches (Bergstra et al., 2011; Snoek et al., 2012; Hutter et al., 2011; Srinivas et al., 2010; Karnin et al., 2013; Li et al., 2017; Falkner et al., 2018; Awad et al., 2021) have significantly advanced this field, and they have been empirically shown to outperform both manual configuration (Hutter et al., 2019; Bischl et al., 2023; Santu et al., 2022; Yang & Shami, 2020) and random search (Bergstra & Bengio, 2012).

HPO is often casted as a black-box optimization problem (BBOP), where the goal is to search for an HP configuration $\boldsymbol{\lambda} \in \boldsymbol{\Lambda} = \Lambda_1 \times \dots \Lambda_n$ with an objective value $\mathcal{L}(\boldsymbol{\lambda})$ as small as possible without any explicit knowledge of the ML loss function $\mathcal{L} : \boldsymbol{\Lambda} \to \mathbb{R}$. Existing methods (see examples above) to this end essentaily comprises 3 key components: $i$): a *search space*, $ii$): an *optimization strategy*, and $iii$): *model evaluation*. While the development of both efficient searching mechanisms and evaluation strategies has received considerable attention in recent years, the intricate interplay between model HPs and predictive losses, which plays a pivotal role in understanding HPO problems, remain notably underexplored. Such lack of knowledge in turn hampers the transparency and explainability (Dwivedi et al., 2023) of HPO solvers, which often function as black-boxes as well. Consequently, this results in limited human trust in HPO methods and hinders their wide-spread application (Drozdal et al., 2020; Bischl et al., 2023) Unfortunately, given the high-dimensional, hybrid nature of HP configuration space, it is far from trivial to open up such black box.

The *fitness landscape* metaphor, which was pioneered by Wright in 1932 in evolutionary biology, has been widely recognized as a powerful tool for analyzing BBOPs in the evolutionary computation community (Malan, 2021). It can be envisioned as a (hyper-)surface as formed by objective values, over the high-dimensional space of possible configurations (Romero et al., 2013). Since the 90s, a plethora of fitness landscape analysis (FLA) methods have been developed to conduct exploratory analysis on landscape characteristics of BBOPs (Zou et al., 2022). Such methods are useful, as they

Figure 1: 2D visualization of HP loss landscapes for: (a-d) `CNN` on 9 HPs and $6,480$ configurations, (e-f) `XGBoost` regressor on 5 HPs and $14,960$ configurations, under different scenarios. Colors indicate ranks of configurations (lower values are better). Coordinates are projected using UMAP.

are able to extract landscape measures that are indicative of problem difficulty and how a certain search mechanism would perform on it (Smith-Miles & Lopes, 2012; Hutter et al., 2014b; Qasem & Prügel-Bennett, 2010). This knowledge would then advance the understanding of the problem characteristics (Huang & Li, 2023), assist the selection and configuration of problem solvers (Kerschke et al., 2019; Schede et al., 2022), navigate the design of new algorithms (Qasem & Prügel-Bennett, 2010), and enhance the explainability and trust for optimization (Thomson et al., 2023).

Recently, the use of FLA in analyzing HP and the related AutoML loss landscapes has also received considerable attention. Various works have studied diverse structural characteristics of these landscapes including neutrality, modality, and fitness distance correlation (e.g., Pushak & Hoos (2022); Teixeira & Pappa (2022); Pimenta et al. (2020); Schneider et al. (2022)). However, such works suffer from limited setups and fail to interrogate the connection between landscape characteristics and the success of HP optimizers, which often run in a wide range of scenarios (e.g., different *models*, *datasets* and *fidelities*). It remains unclear whether the HP loss landscapes induced on different settings share certain characteristics or patterns, and how such commonalities could be potentially exploited by HP optimizers. On the other hand, we argue that existing analytical methods are insufficient to provide a comprehensive analysis and comparision on HP loss landscapes since:

☞ The ability to visualize landscapes is crucial for enabling intuitive understandings of their complex topologies (Michalak, 2019). However, HP loss landscapes are notoriously difficult to visualize in a human-comprehensible fashion due to their high-dimensional nature. Some existing methods address this problem by plotting only one or two HPs each time (e.g., Friedman (2001); Akiba et al. (2019)), which fail to provide an integrated view of the global landscape structure. Other works applied dimensionality reduction techniques to project the landscape into 2D space (e.g., Michalak (2019); Biedenkapp et al. (2018); Walter et al. (2022)), but the resulting plot is not able to preserve the overall topography as well as neighborhood structure of the landscape.

☞ There is no tangible method for quantifying the similarity between different HP loss landscapes. Despite *general* FLA metrics could already capture informative landscape characteristics, practices in automated algorithm selection demonstrate that *domain-specific* metrics are also crucial as a complementary source of information for better characterizing the target problem (Smith-Miles, 2008; Smith-Miles & Lopes, 2012). However, none of the prior works have considered such perspectives when comparing HP loss landscapes.

The overarching goal of this paper is to gain an integral view of the HP loss landscapes induced on different scenarios and thereby provide new insights to the community. To this end, we develop a dedicated landscape analysis framework to enable comprehensive analysis and comparisons among HP loss landscapes. It incorporates ❶: a novel neighborhood-aware HP loss landscape *visualization* method applicable to high-dimensions, ❷: a series of FLA metrics quantifying landscape *structural characteristics*, and ❸: 3 similarity metrics that leverage rankings of HP configurations to allow for informative landscape *similarity* quantification in the HPO context. Through empirical analysis on $1,500$ landscapes across $6$ ML models and $67$ datasets with more than $11$ million configurations, we are ambitious to advance the understanding of the following four fundamental HPO scenarios:

**HP Landscapes of Test Loss Versus Training Loss.** 'Overfitting' is one of the biggest interests and concerns in the ML community (Ng, 1997; Caruana et al., 2000; Recht et al., 2019; Belkin et al., 2018; Roelofs et al., 2019; Ishida et al., 2020). However, there is a lack of in-depth understanding on how test loss correlates with training loss across a broad HP landscape, and what specific properties distinguish regions that generalize well from poorly generalized ones. In this paper, by using our

fitness landscape analysis framework, we find that the test loss landscapes resemble their training counterparts in terms of both structural characteristics and performance rankings (see, e.g., Figure 1 (a) versus (b)), and configurations with small training error are likely to achieve a mild generalization error. However, significant discrepancies can also occur (see, e.g., Figure 1 (e) versus (f)) depending on both the choice of certain HP combinations and the dataset at hand. In such cases, struggling to reduce the training loss has little or even negative effect to refining the generalization loss.

**HP Loss Landscapes Across Fidelities.** Given the time-demanding nature of model evaluation, multi-fidelity HPO methods (Karnin et al., 2013; Kandasamy et al., 2016; Li et al., 2017; Kandasamy et al., 2017; Falkner et al., 2018; Awad et al., 2021) have achieved prominent performance by more efficient resource allocation. However, the validity of their underpinned assumption, i.e., the ranking of configuration performance would stay close to the ground truth under fidelities (Bischl et al., 2023), remains unclear (Pushak & Hoos, 2022). Our empirical results are highly inspiring to support such assumption and show that landscapes with lower fidelities are highly consistent with full-fidelity landscapes w.r.t. both structural characteristics and performance ranks (Figure 1 (c)).

**HP Loss Landscapes Across Datasets.** Leveraging priors obtained from previous tasks to expedite the learning process for new tasks is another promising direction for HPO (Feurer et al., 2015b; Bardenet et al., 2013; Wistuba et al., 2015b; Kim et al., 2017; Rakotoarison et al., 2022; Wistuba et al., 2015c; Vanschoren, 2018; Swersky et al., 2013). These approaches are grounded in the hypothesis that 'knowledge' about HPO landscapes—such as configuration quality, hyperparameter importance and their interactions (Hutter et al., 2014a; Watanabe et al., 2023b; Probst et al., 2019; van Rijn & Hutter, 2017)—is correlated across related datasets when defined under a certain distance measure. A natural question that arises is whether this knowledge remains consistent when applied to a broader range of tasks. Our results on a diverse set of 67 datasets show that performance rankings, HP importance and their interactions, are largely shared across tasks (Figure 1 (d)).

**HP Loss Landscapes Across Models.** Methods rooted in Bayesian optimization (e.g., Snoek et al. (2012); Bergstra et al. (2011); Feurer et al. (2015a); Kandasamy et al. (2017)) and search space pruning (e.g., Wistuba et al. (2015a); Perrone & Shen (2019); Wang et al. (2020)) implicitly assume that the quality of configurations is locally correlated rather than a full randomness. This is also true for meta-heuristic methods (e.g., Friedrichs & Igel (2005); Lessmann et al. (2005); Cawley (2001); Guo et al. (2008)), which are even more dependent on specific landscape properties. While it may seem intuitive that HP loss landscapes would differ depending on the target ML model, in practice the fact is often that common HPO methods perform robustly for different models. This implies that, despite superficial differences, the general family of HP loss landscapes may share certain inherent patterns/properties. We verified this hypothesis by synthesizing the results from diverse FLA metrics characterizing HP loss landscape geometry combined with visual inspections (see, e.g., Figure 1 (a, e)). The results gathered from 1, 500 landscapes of 6 ML models under different scenarios, reveal a universal picture of the HP loss landscapes. In this picture, HP loss landscapes are smooth, nearly unimodal, containing a large number of neutral regions; configurations with similar performance are locally clustered; the landscape becomes flatter around the optimum configuration.

## 2 HPO LANDSCAPE ANALYSIS METHODS

This section introduces our analytical framework developed to enable exploratory analysis on different HP loss landscapes and perform comparisons between them. Due to the page limit, we only provide a brief sketch of these methods here while more detailed discussions are in Appendix B.

**HP Loss Landscape Construction.** The HP loss landscape can be formulated as a triplet $\langle \mathbf{\Lambda}, \mathcal{L}, \mathcal{N} \rangle$ with three ingredients: $i$) a search space $\mathbf{\Lambda}$ of feasible configurations that consists of pre-evaluated, discretized grid values (see Appendix F.1), $ii$) a ML loss function $\mathcal{L} : \boldsymbol{\lambda} \to \mathbb{R}$, and $iii$) a neighborhood structure $\mathcal{N}$ that specifies which configurations are neighbors to each other. Note that the form of $\mathcal{N}$ depends on a distance function $\mathrm{d} : \boldsymbol{\lambda} \times \boldsymbol{\lambda} \to \mathbb{N}$. Following Pushak & Hoos (2022), we define all categorical values of a HP to be distance 1 from each other (i.e., the values are non-ordinal). For a numerical HP, we define the distance between two values to be the number of steps between them on the grid used for discretization. Such distance measure is able to mimic the tunning strategy of human experts when combined with elaborately designed grid values. Based on this, the total distance between two configurations $\boldsymbol{\lambda}_i$ and $\boldsymbol{\lambda}_j$ is then sum of the distances between the respective pairs of HP values, and we say they are neighbors to each other (i.e., $\boldsymbol{\lambda}_j \in \mathcal{N}(\boldsymbol{\lambda}_i)$), if $\mathrm{d}(\boldsymbol{\lambda}_j, \boldsymbol{\lambda}_i) = 1$. Fi-

Table 1: Summary of the FLA metrics used in our landscape analysis framework.

| METRICS | SYMBOL | DOMAIN | WHAT A HIGHER VALUE IMPLIES |
|---|---|---|---|
| Performance Assortativity[1] | $\mathcal{L}$-ast | $[-1, 1]$ | HP Configurations with similar $\mathcal{L}$ values are more likely to be neighbors to each other. |
| Autocorrelation[2] | $\rho_a$ | $[-1, 1]$ | The landscape is smoother |
| Neutrality Distance Correlation | NDC | $[-1, 1]$ | The landscape is more likely to be flatter near the optimum. |
| Mean Neutrality[3] | $\bar{\nu}$ | $[0, 1]$ | There are many 'plateaus' in the landscape. |
| No. Local Optima | $n_{lo}$ | $\mathbb{N}^+$ | There are many 'valleys' or 'peaks' in the landscape. |
| Mean Basin Size | $\bar{s}_{\mathcal{B}}$ | $\mathbb{R}^+$ | The local optima are hard to be escaped from. |

[1]Newman (2010); [2]Weinberger (1990), [3]Reidys & Stadler (2001)

nally, the HPO landscape is constructed as a directed graph where the vertices are HP configurations and an improving edge $e_{i,j} \in \mathcal{E}$ is traced from $\boldsymbol{\lambda}_i$ to $\boldsymbol{\lambda}_j$ if $\boldsymbol{\lambda}_j \in \mathcal{N}(\boldsymbol{\lambda}_i)$ and $\mathcal{L}(\boldsymbol{\lambda}_j) < \mathcal{L}(\boldsymbol{\lambda}_i)$. We say that a configuration $\boldsymbol{\lambda}_\ell$ is a local optimum if $\forall \boldsymbol{\lambda}' \in \mathcal{N}(\boldsymbol{\lambda}_\ell)$, we have $\mathcal{L}(\boldsymbol{\lambda}_\ell) < \boldsymbol{\lambda}'$. In addition, we say that $\boldsymbol{\lambda}_j$ is a *neutral* neighbor of $\boldsymbol{\lambda}_i$ if their performance difference is negligible ($\leq 1‰$).

**Landscape Visualization.** We develop a first-of-its-kind, highly interpretable method for visualizing the topography of high-dimensional HP loss landscapes by leveraging graph representation learning (Hamilton, 2020) combined with dimensionality reduction (Draganov et al., 2023) techniques. Specifically, we first extracted low-dimensional features for each node in the graph. To this end, we use HOPE (Ou et al., 2016) node embedding method because it could preserve high-order proximities between configurations. Then, we compress the obtained feature vectors into 2 components using the UMAP (McInnes & Healy, 2018) algorithm, and thus allowing configurations to be laid out in 2D scatter plot. To further refine the interpretability of the obtained plots, we additionally apply a linear interpolation and thereby generate a smooth landscape surface.

**Quantifying Landscape Characteristics.** To quantitatively assess the structural characteristics of HP loss landscapes, we employ a series of dedicated FLA metrics summarized in Table 1 as surrogate features. There are many other metrics for characterizing landscape properties (see Zou et al. (2022) for a detailed review), but our selected ones are particularly useful for this study as they cover the most essential landscape properties (i.e., modality, neutrality and smoothness) that are related to algorithm behaviors. More importantly, they are intuitive enough even for non-experts in FLA.

**Landscape Similarity in Terms of Performance Ranking.** The comparison of rankings of HP configurations' performance is the essence of a large corpora of HPO methods (Hutter et al., 2019). We thereby ground our landscape similarity measure of HP loss landscapes on the consistency of their performance ranks, denoted as $R(\mathcal{L}(\boldsymbol{\lambda}))$, to allow more informative results in the HPO context. Specifically, we use 3 statistical metrics with complementary perspectives: 1) *Spearman's* $\rho_s$, it measures the association of the performance ranks of configurations in two landscapes (Spearman, 1961), 2) Kaggle's *Shake-up* metric (Trotman, 2019), it assesses the average movement of configuration rankings across two landscapes. 3) The $\gamma$-*set similarity* (Watanabe et al., 2023a), it quantifies the ratio of overlaps between top-10% regions of two landscapes divided by their unions.

In addition to these, to investigate the consistency of HP importance and interaction under different scenarios, We apply the widely used functional ANOVA method (Hutter et al., 2014a) to assess the variance contribution of every HP $\lambda \in \boldsymbol{\lambda}$ as well as their interactions up to the 3$^{rd}$ order.

## 3 EXPERIMENTAL SETUP

Table 2 summarizes the meta-information of our empirical study, while detailed HP search space of each model and the principles we follow in designing them, are left in Appendix F.1. We first consider decision tree (DT) (Safavian & Landgrebe, 1991) and three types of its ensembles: random forest (RF) (Breiman, 2001), XGBoost (Chen & Guestrin, 2016) and LightGBM (Ke et al., 2017). We analyze the HP space of these models using the tabular benchmark proposed in Grinsztajn et al. (2022), which comprises 25 regression and 32 classification tasks (see Appendix F.2). These datasets span a broad range of complexities in terms of number of instances and features and are thus idea choice for comprehensive inspection of landscape characteristics. In addition to these, we also study convolutional neural networks (CNNs) (Krizhevsky et al., 2012) on six classic image classification

Table 2: Summarization of meta-information of our empirical study.

| MODELS | | | DATASETS | | | | | FIDELITIES | | SUMMARIZATION | |
|---|---|---|---|---|---|---|---|---|---|---|---|
| Model | Total HPs | Total Configs. | Cat. Class. | Cat. Reg. | Num. Class. | Num. Reg. | Image Class. | Training Data | Training Epochs | Total Configs. | # Land- -scapes |
| XGB | 5 | 14,960 | 15 | 7 | 17 | 18 | - | $\{0.1, 0.25, \text{all}\}$ | - | 2.56M | 342 |
| RF | 6 | 11,250 | 15 | 7 | 17 | 18 | - | $\{0.1, 0.25, \text{all}\}$ | - | 1.92M | 342 |
| LGBM | 5 | 13,440 | 15 | 7 | 17 | 18 | - | $\{0.1, 0.25, \text{all}\}$ | - | 2.30M | 342 |
| DT | 5 | 24,200 | 15 | 7 | 17 | 18 | - | $\{0.1, 0.25, \text{all}\}$ | - | 4.14M | 342 |
| CNN | 8 | 6,480 | - | - | - | - | 6 | $\{0.1, 0.25, \text{all}\}$ | $\{10, 25, 50\}$ | 0.35M | 108 |
| FCNet | 9 | 62,208 | - | - | - | 4 | - | - | $\{10, 50, 100\}$ | 0.19M | 24 |
| Total (Before accounting 5-fold cross-validation): | | | | | | | | | | 11.15M | 1,500 |

tasks (Appendix F.2) using a joint architecture and hyperparameter search (JAHS) (Bansal et al., 2022) space. We additionally consider another JAHS scenario, for which we adopt the NASBench-HPO (Klein & Hutter, 2019) data included in HPOBench (Eggensperger et al., 2021). This includes 62,208 configurations of a feed-forward neural network (FCNet) evaluated on 4 UCI datasets

For each dataset, unless predefined, we randomly split the data into training (80%) and test (20%) set. For all HP configurations $\boldsymbol{\lambda} \in \Lambda$ of each model, we exhaustively evaluate $\mathcal{L}(\boldsymbol{\lambda})_{\text{train}}$ and $\mathcal{L}(\boldsymbol{\lambda})_{\text{test}}$ using 5-fold cross-validation. Here, we use root mean squared error (RMSE) and $R^2$ score to serve as the loss function $\mathcal{L}$ for regression tasks, and for classification, we use accuracy and ROC-AUC. We control the fidelity of the training by varying the number of training instances to $\{10\%, 25\%, 100\%\}$ of the whole training data. For CNN, we additionally set the budget for the number of epochs to $\{10, 25, 50\}$ and thus obtain a total of $3 \times 3$ different levels of fidelity. For FCNet, we vary fidelity by using meta-data at the $\{10, 50, 100\}$-th epoch. At the end, we obtain a total of $1,500$ landscapes with more than 11M distinct HP configurations. To further demonstrate the transferability and potential impact of our proposed landscape analysis framework, we also employ it to analyze NASBench-101 (Ying et al., 2019), a well-known neural architecture search (NAS) benchmark.

## 4 RESULTS AND ANALYSIS

In this section, we seek to investigate HP loss landscapes under the four scenarios posed in Section 1. We will start from providing an universal view of the general characteristics of HP loss landscapes of different ML models (Section 4.1). We then explore and compare landscapes under: $i$) training and test setups (Section 4.2), $ii$) different fidelities (Section 4.3), $iii$) different datasets (Section 4.4).

### 4.1 OVERALL CHARACTERISTICS OF HP LOSS LANDSCAPE OF ML MODELS

From landscape visualizations depicted in Figure 2 (a), we have a general impression that HP loss landscapes of ML models are highly structured and share certain patterns: *they are relatively smooth; configurations are clustered in terms of performance; there is a highly distinguishable plateau consisting of prominent configurations, where the terrain becomes flatter.* This impression is consistent with the FLA metrics reported in Figure 3, from which we see that landscapes for all models are:

**Fairly smooth and clustered.** The high $\mathcal{L}$-ast and $\rho_a$ values for $\mathcal{L}_{\text{test}}$ landscapes shown in Figure 3 (a) and (b) respectively imply that configurations with similar $\mathcal{L}_{\text{test}}(\boldsymbol{\lambda})$ tend to be locally connected, where a small change in $\boldsymbol{\lambda}$ is not likely to cause dramatic variation of $\mathcal{L}_{\text{test}}(\boldsymbol{\lambda})$. This observation is similar to the findings in reinforcement learning (RL) (Eimer et al., 2023), where the transitions between different parts of the HP landscapes of RL are also found to be quite smooth. This property makes the HP landscapes favorable to Bayesian optimization and search space pruning techniques, as it would be easier to separate the promising regions from the poorly performing ones. On the other hand, if the landscape is rugged instead, in which $\mathcal{L}_{\text{test}}(\boldsymbol{\lambda})$ of very different levels often mixed together, it would become more difficult for such techniques to identify a clear promising region.

**Nearly unimodal.** As shown in Figure 3 (e), we find that a considerable fraction of the $\mathcal{L}_{\text{test}}$ landscapes are *unimodal*, whereas for other landscapes, there could be a handful to dozens (DT) of local optima. This is somewhat contradictory to Pushak & Hoos (2022) at the first thought, in which the authors found that almost all landscapes they studied are nearly unimodal. However, when taking a closer look at the local optima in our landscapes, we find that they usually feature a small basin of

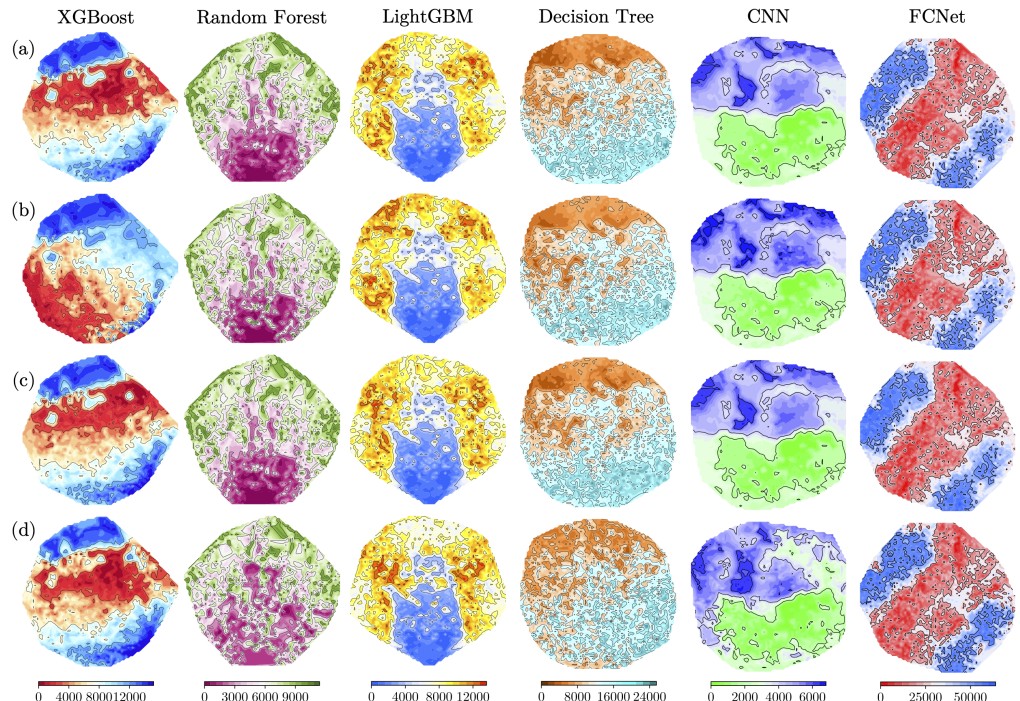

Figure 2: 2D visualization of HP loss landscapes of 6 ML models under different scenarios: **(a)** $\mathcal{L}_{\text{test}}$ landscape on baseline datasets (44059 for tree-based models, CIFAR-10 for `CNN`, protein structure for `FCNet`), **(b)** $\mathcal{L}_{\text{train}}$ landscape on baseline datasets, **(c)** Low-fidelity $\mathcal{L}_{\text{test}}$ landscape on baseline datasets, **(d)** $\mathcal{L}_{\text{test}}$ landscape on different datasets (44143 for tree-based models, Fashion MINIST for `CNN`, slice localization for `FCNet`). Colors indicate $R(\mathcal{L})$ (lower rank values are better).

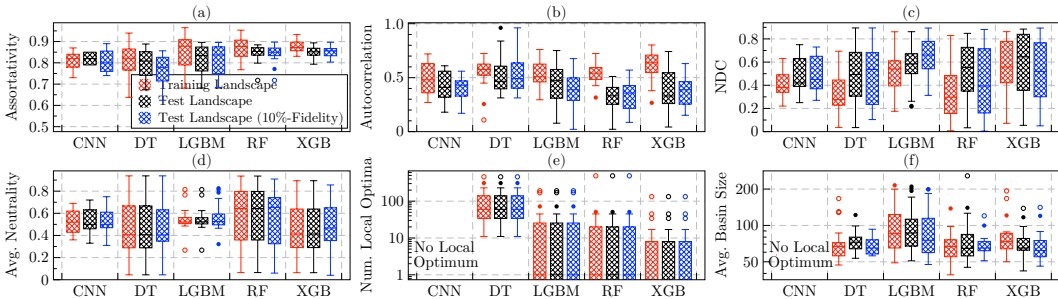

Figure 3: Distribution of FLA metrics introduced in Table 1 for each model across all datasets for landscapes of 1) $\mathcal{L}_{\text{test}}$, 2) $\mathcal{L}_{\text{train}}$ and 3) $\mathcal{L}_{\text{test}_{\text{LF}}}$.

attraction (Figure 3 (f)). This makes them relatively 'shallow', and thus would not pose significant obstacles to optimization. However, beyond the results in Figure 3, we find that `FCNet` landscapes on the 4 UCI datasets possess 24 to 347 local optima, with $\bar{s}_{\mathcal{B}}$ up to $2,372$ (Appendix D), implying a strong attraction for optimizers. Pushak & Hoos have also reported similar observations on these four landscapes, and they speculated the reason could be that these scenarios fall into the over-parameterized regime. While we agree with this reasoning, we seek to conduct further analysis on the local optima using the local optima network (LON) (Ochoa et al. (2008), Appendix B.3). We find that despite the pressence of many other local optima, the global optima still plays a pivotal role in the connectivity of the LON (Appendix D). Therefore, many local optima can eventually escape the global optimum via cetain strategies (e.g., a perturbation), though this may take additional efforts.

**Highly neutral; planar around the optimum.** As depicted in Figure 3 (d), we can clearly see that HP loss landscapes are often featured in high neutrality. This indicates that a large portion of 1-bit

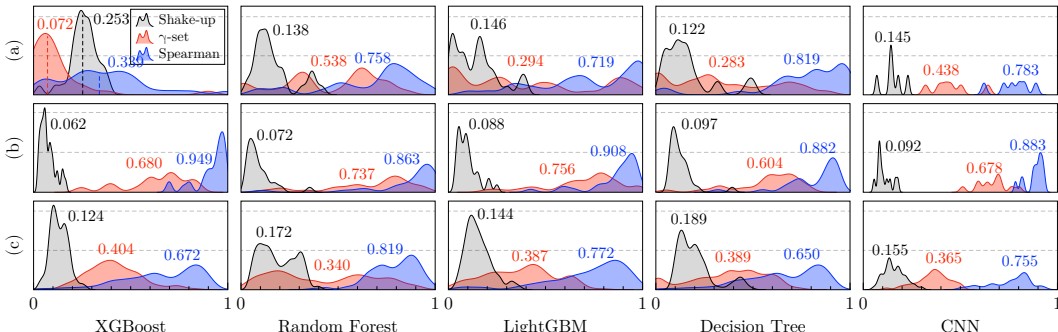

Figure 4: Distribution of Spearman, Shake-up and $\gamma$-set metrics between **(a)** $\mathcal{L}_{\text{test}}$ and $\mathcal{L}_{\text{train}}$, **(b)** $\mathcal{L}_{\text{test}}$ and $\mathcal{L}_{\text{test}_{\text{LF}}}$, **(c)** $\mathcal{L}_{\text{test}}$ across datasets. Medians are labeled beside each plot.

moves in the landscape will result in subtle change in $\mathcal{L}_{\text{test}}(\boldsymbol{\lambda})$ (i.e., $\leq 1‰$). We postulate a major reason for this is the *low effective dimensionality* (Bergstra & Bengio, 2012) of HPO problems: usually only a small subset of all available HPs have obvious influence on performance. Despite landscape neutrality can largely vary with the choice on which HPs to analyze and their respective values, considering the fact that we have ready removed totally unimportant HPs from teh search space, moves with subtle performance shifts can actually be more prevalent than one may expect. Such phenomenon is more pronounced for the well-performing regions, as illustrated by the high NDC values in Figure 3 (c). It suggests that as we move closer to the global optimum, it is more likely to encounter neutral moves, and the landscape becomes *flatter*. This is in line with Probst & Boulesteix (2017); Pimenta et al. (2020) and practical experience: the gain of tuning HPs usually progressively decreases as approaching the best reachable performance. Such property also poses challenges to optimizers, as there is little gradient information that can be utilized for navigation towards fitter configurations (Muñoz et al., 2015).

Overall, despite certain exceptions, we see that the family of HP loss landscapes tend to share various high-level properties, whereas the exact topologies would vary with models. This explains why in practice, one can often expect an optimizer to work relatively robustly across a wide range of scenarios. In addition, most properties here also seem to generalize the NAS problems (Appendix C), except that we find the NAS landscapes tend to have lower neutrality and more local optima.

## 4.2 TRAINING AND TEST HPO LANDSCAPES

Figure 2 (a) and (b) provide a visual comparison of $\mathcal{L}_{\text{train}}$ and $\mathcal{L}_{\text{test}}$ landscapes for different models, respectively. We could infer from the plots that the *structural characteristics* of the $\mathcal{L}_{\text{train}}$ landscapes highly our previously discussed properties. On the other hand, for the *performance rankings*, we notice that $\mathcal{L}_{\text{train}}$ generally correlate with $\mathcal{L}_{\text{test}}$ for RF, DT, LGBM and CNN, whereas for XGBoost, there is significant shifts in performance between the two cases. We further quantitatively verify such observations using the FLA metrics and the landscape similarity metrics introduced in Section 2.

**Structural characteristics.** From Figure 3, we could clearly see that the landscape characteristics for $\mathcal{L}_{\text{train}}$ and $\mathcal{L}_{\text{test}}$ are highly consistent for most studied scenarios. More specifically, it is surprising to see that $\mathcal{L}_{\text{train}}$ landscapes tend to yield relatively higher $\mathcal{L}$-ast and $\rho_a$, suggesting a smoother and more structured terrain. Meanwhile, the NDC values are lower in the training scenarios. These observations imply that $\mathcal{L}_{\text{train}}$ landscapes are even more benign than $\mathcal{L}_{\text{test}}$ landscapes. In addition, we find that $\bar{\nu}$, $n_{\text{lo}}$ and $\bar{s}_{\mathcal{B}}$ values rarely change between $\mathcal{L}_{\text{train}}$ and $\mathcal{L}_{\text{test}}$ landscapes. Notably, local optima found in $\mathcal{L}_{\text{train}}$ and $\mathcal{L}_{\text{test}}$ landscapes are almost (if not all) identical. These indicate that the relative performance in local neighborhoods tend to remain similar in the two cases, despite the variations in their numerical values and the global performance rankings.

**Landscape similarity in terms of performance rankings.** We quantified the similarity between all pairs of $\mathcal{L}_{\text{train}}$ and $\mathcal{L}_{\text{test}}$ landscapes for all 5 models using the three metrics introduced in Section 2 as shown in Figure 4 (a). Overall, we observe that $R(\mathcal{L}_{\text{train}})$ and $R(\mathcal{L}_{\text{test}})$ are globally correlated for all models except XGBoost, as indicated by the significant $\rho_s$ values (median $> 0.7$) and low Shake-up metrics (median $< 0.15$). However, when zooming into the top-10% regions, we find that

Figure 5: (a) Scatter plot of $\mathcal{L}_{\text{train}}$ versus $\mathcal{L}_{\text{test}}$ for all $14,960$ configurations of `XGBoost` on the dataset #44059. $\boldsymbol{\lambda}^*_{\text{test}}$ is marked by red star ☆, (b-e) The same plot with colormap added according to HPs values. Warmer color indicate higher values.

the majority of our studied scenarios reveal low $\gamma$-set similarities. It indicates that the generalization gap is larger in prominent regions where configurations are highly adapted to the training set. This phenomenon is more severe for `XGBoost`, where the median $\gamma$-set similarity is only $0.07$, and there is also a poor $\rho_s$ value (median $= 0.34$) and high Shake-up score (median $= 0.25$).

In order to gain more insight into such generalization gaps for `XGBoost`, we create scatter plots of $\mathcal{L}_{\text{test}}$ versus $\mathcal{L}_{\text{train}}$ on dataset #44059 as shown in Figure 5 (a). We decompose the pattern into two modes: During the first mode, $\mathcal{L}_{\text{test}}$ highly correlates with $\mathcal{L}_{\text{train}}$ as it decreases, and the models in this stage *underfit* the data. In the next mode, as points struggle to further move on the $x$-axis ($\mathcal{L}_{\text{train}}$), they stagnate or even significantly increase on the $y$-axis ($\mathcal{L}_{\text{test}}$), indicating strong evidence of overfitting. In particular, we can see a plateauing trend near the $x$-axis, where some models overly excel on the training data, but performing poorly on the test set.

To further investigate which kinds of configurations are likely to lead to overfitting, we color the points with respect to their HP values as shown in Figure 5 (b-e). We are excited to see that the generated plots demonstrate clear patterns between the value of each HP and the resulted performance. In particular, we find that learning rate, max depth and subsample have significant impact on $\Delta\mathcal{L}$. However, the generalizability of a learner is not monopolized by a single one of them; instead, it depends on their cumulative interactions. For example, the largest $\Delta\mathcal{L}$s are observed for learners that features a large learning rate, deep base trees, combined with low subsample rate, but any of these HP settings alone does not necessarily lead to the worst case performance. In addition to this, we notice that such generalization gap is also related to dataset characteristics and weakly correlated across models, and we dissuss more about this matter in Appendix E.

### 4.3 HPO Landscapes with Different Fidelities

Figure 2 (c) shows the low-fidelity test loss landscapes (denoted as $\mathcal{L}_{\text{test}_{\text{LF}}}$) for each model (using 10 epochs for `FCNet`, and 10% training data for others). From the plots, we could see that $\mathcal{L}_{\text{test}_{\text{LF}}}$ landscapes are highly consistent with $\mathcal{L}_{\text{test}}$ in terms of both structural characteristics and performance rankings. More specifically, as reflected in Figure 3, all measured FLA metrics of $\mathcal{L}_{\text{test}_{\text{LF}}}$ landscapes showed little difference compared to $\mathcal{L}_{\text{test}}$ landscapes across all studied scenarios. For performance rankings, Figure 4 (b) depicts the distribution of the 3 similarity indicators between $\mathcal{L}_{\text{test}_{\text{LF}}}$ and $\mathcal{L}_{\text{test}}$ across all datasets for each model. We could observe a high Spearman correlation (median $> 0.85$) between $\mathcal{L}_{\text{test}}$ and $\mathcal{L}_{\text{test}_{\text{LF}}}$ for all models, and the $\gamma$-set similarities between the top-10% configurations are also prominent, with medians larger than 60%. These imply that $R(\mathcal{L}_{\text{test}})$ and $R(\mathcal{L}_{\text{test}_{\text{LF}}})$ are highly consistent for the majority of our studied scenarios and there is large overlap between the promising regions of the two landscapes. In addition, the Shake-up scores yield low values (median $< 0.1$), suggesting that on average the difference between $R(\mathcal{L}_{\text{test}})$ and $R(\mathcal{L}_{\text{test}_{\text{LF}}})$ is less than 10%. Additional results on `FCNet`, NAS-Bench-101 in Appendix D and Appendix C respectively are also consistent with our findings here.

### 4.4 HPO Landscapes Across Datasets

Figure 2 (d) shows the $\mathcal{L}_{\text{test}}$ landscapes for each model on a different dataset. From the figure, it is exciting to see that the high-level topography of HP loss landscape are preserved when transferring to a new task. In particular, we find that the top regions in the original landscape generally retain

their positions, despite changes in their exact contours. The FLA metrics we previously saw in Figure 3 support such observation, from which we have been able to draw an unified picture of the characteristics of HP loss landscapes. In addition, from the similarity metrics reported in Figure 3, we can infer that the measured performance reveal clear Spearman correlations (median $> 0.65$) across datasets. More importantly, the overlap between well-performing regions, as indicated by the $\gamma$-set similarity, also achieves medians around $40\%$. In addition, it is intriguing to find that despite the dataset #45041 (9K instances and 255 features) and #45047 (1M instances and 5 features) seem to be totally different, they reveal $\rho_s > 0.7$ and $\gamma$-set similarity $> 40\%$ for all 4 tree-based models.

In addition to performance rankings, Figure 6 illustrates the contribution of each HP and their interactions to model performance assessed by the functional ANOVA method. The results indicate that some (combination of) HPs are typically important for many datasets for a given model. For example, learning rate consistently contributes a large portion of variance to model performance for `LightGBM`, and its interactions with the number of leaves and estimators are also important. These observations are similar with van Rijn & Hutter (2018), which find also conclude that certain HPs of a ML model are important for a wide spectrum of datasets by analyzing meta-data from OpenML platform.

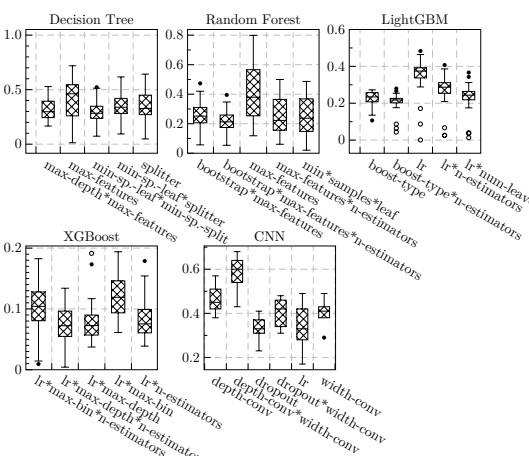

Figure 6: Importance of top-5 HPs as determined by functional ANOVA method for each model.

As discussed in Section 4.1, HP loss landscapes often involve a large number of none-improvement moves, especially near the optimum. We also see that there is clear division between the promising regions and the poorly-performaning ones. Therefore, leveraging prior knowledge from previously tasks should be able to greatly expedite the searching process by means like warm-starting HPO from good configurations or carefully selecting candidate HPs and crafting the search space. More importantly, based on our results, we note that this should not be only limited to *similar* tasks defined under certain rules, since they may not always available. On the other hand, seemingly different tasks could still provide informative information as we see above. Our additional results for `FCNet` on 4 datasets, and NAS-Bench-201 across CIFAR-10/100 as well as ImageNet datasets (Appendix C), also revealed similar highly-transferable conclusions.

## 5 DISUCCSIONS AND CONCLUSIONS

By conducting large-scale exploratory analysis on $1,500$ HP landscapes of 6 ML models with over 11M model configurations under different scenarios, this paper reveals an unified portrait of their topographies in terms of smoothness, neutrality and modality. We also show that these properties are highly transferable across datasets and fidelities, and thus provide fundamental evidence to support the effectiveness of transfer and multi-fidelity methods, which in privious practices, is mainly based on intuition. However, while our findings are observed for the majority studied scenarios, we do observe some exceptions. For example, most landscapes inspected reveal a nearly unimodal structure, but some of them can have dozens to a few hundreds of local optima with non-negligible basin sizes (e.g., `FCNet`). Also, there are cases when landscapes under lower fidelities or on a different task reveal very different patterns, as shown by the long tails of the similarity distributions in Figure 4. Further explorations by interrogating the relationship between dataset characteristics may provide a even more comprehensive understanding of the HPO landscape.

Our developed FLA framework in this work, has shown great potential for enabling both qualitative and quantitative understanding for a wider range of AutoML problems. While it currently relies on large-scale, pre-evaluated data points for landscape construction, we think it can be a promising direction to integrate it with existing AutoML frameworks, and thus allow for on-the-fly analysis of problem landscapes, thereby allowing it to be accessible by a broader range of stakeholders.

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

## A  BACKGROUND AND RELATED WORK

**Fitness Landscape of BBOPs.** A central topic in optimization concerns how the intricate interplay of *decision variables* determine the targeted *objective values*. For example, geneticists and biologists are interested in the relationship between genotypes and the phenotypes and functions of organisms that may affect their evolutionary success (De Visser & Krug, 2014). In chemical physics, scientists seek to explore how the energy of a system is dependent on its various configurational or conformational states (Brooks III et al., 2001). The *fitness landscape* can be envisioned as a surface over the high-dimensional space as formed by decision variables, where the objective values are represented as the elevation of the landscape. The structure of this surface describes the spectrum of possible objectives values across the variable space and thus strongly influences the optimization. In addition, searching trajectories of problem solvers can be thought as strategic walks on the corresponding problem landscape. Successful applications of this metaphor to analyze black-box systems have advanced the understanding of many fields (He & Liu, 2016; Puchta et al., 2016; Shires & Pickard, 2021; Brooks III et al., 2001). Most related to our work is the fitness landscape analysis (FLA) for black-box optimization problems (BBOPs) (Malan, 2021). Many metrics that quantify the structural characteristics of fitness landscapes have been developed over the years to describe the topography of BBOP landscapes, e.g., autocorrelation, neutrality, modality, epistasis variance, evolvability, etc.

**Fitness Landscape on other ML Systems.** In addition to the works that focus on analyzing HP and AutoML landscapes introduced in Section 1, there are also works which applied FLA to study other ML systems, including the neural architecture search (NAS) spaces (Rodrigues et al., 2020; Traoré et al., 2021) and the HP landscapes of reinforcement learning (RL) (Eimer et al., 2023; Mohan et al., 2023). We note that these works are somewhat orthogonal to ours, as we focus on the HP landscapes of ML algorithms. In addition, there is also another line of work on investigating the *loss landscapes* of neural networks (Rakitianskaia et al., 2016; Rodrigues et al., 2022; van Aardt et al., 2017). These works, though seemingly similar, are focused on the neural network losses which are the objectives of weight optimizers during model training, and are thus significantly different from ours.

**HPO Landscape Visualization.** The ability to visualize HP loss landscapes is crucial for enabling intuitive understandings of their complex topologies, which is however, far from trivial in practice due to their high-dimensional nature. Current study methods usually bypass this issue by exmaining only two HPs each time (see, e.g., Akiba et al. (2019); Ørebæk & Geitle (2021)). However, such techqniues can not provide an unified illustration of the total interactive effects of all interested HPs. In the evolutionary computation, there are have been successful attempts on applying dimensionality reduction techniques to visualize the fitness landscape of BBOPs (Michalak, 2019; Huang & Li, 2023; Sass et al., 2022; Walter et al., 2022). Similar work has also been done in the domain of chemical physics (Shires & Pickard, 2021), where UMAP and $t$-SNE are used to visualize the so-called energy landscape.

**Overfitting in Machine Learning.** To empirically investigate the presence of overfitting in real-world ML applications, Roelofs et al. (2019) analyzed massive submission meta-data on the Kaggle platform, and their results suggest little evidence of overfitting due to testset reuse. However, since the Kaggle data is constituted by a diverse set of ML models with very different HP configurations that are not comparable with each other, it prohibits further analysis on which HP configurations are more likely to lead to overfitting.

**Hyperparameter Importance and Interaction.** Understanding which HPs influence model performance to what extend can provide valuable insights into the tuning strategy (Probst et al., 2019). To analyze importance of hyperparameters, one could either use models that are inherently interpretable, e.g., decision trees (Quinlan, 1986), or apply model-agnostic methods such as functional ANOVA (Hutter et al., 2014a; Watanabe et al., 2023b). Built on this technique, van Rijn & Hutter (2017) compared the performance of a variety of algorithms on large set of OpenML datasets and stated that the same hyperparameters were typically important across datasets.

## B  DETAILS OF THE LANDSCAPE ANALYSIS FRAMEWORK

Figure 7 illustrates the conceptual workflow of our proposed HP loss landscapes analysis framework in Section 2. From a high level, our landscape visualizations (Appendix B.1) seek to provide a general sketch of the landscape topography. The observed patterns can then be quantitatively verified

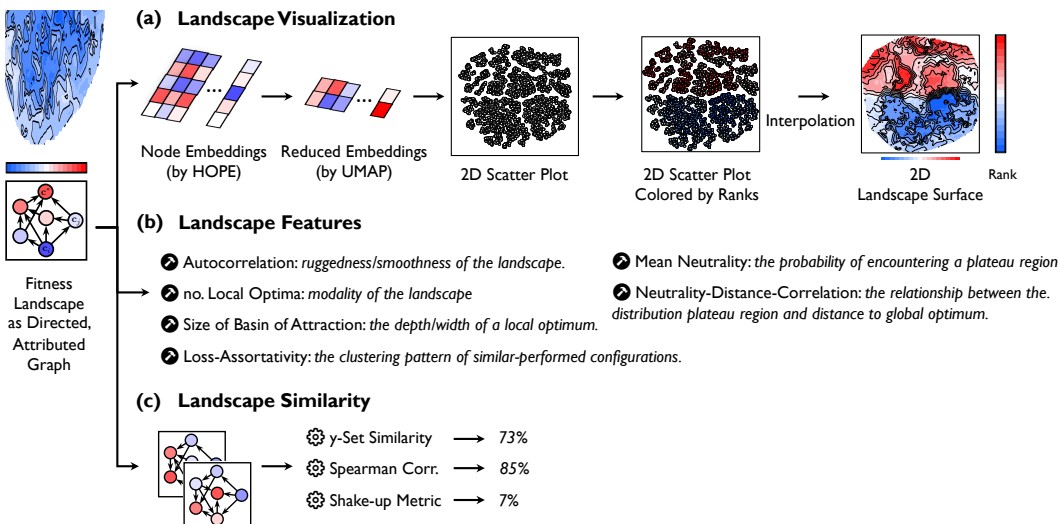

Figure 7: High-level conceptual workflow of our proposed HP loss landscape analysis framework.

via a set of dedicated FLA metrics characterizing complementary aspects of landscape properties (Appendix B.2), while landscapes visualizations, at the same time, can assist the intuitive interpretation of the numerical figures. In addition to these, we also apply 3 dedicated similarity measures to quantify the consistency of configuration performance across landscapes (Appendix B.4).

## B.1 LANDSCAPE VISUALIZATION METHOD

**HOPE Node Embedding.** To preserve the intrinsic neighborhood relationship of HP configurations, our proposed landscape visualization method first needs to learn a low-dimensional embedding for each configuration in the landscape (i.e., each node in the graph). In this paper, we choose HOPE[1] node embedding method to serve this purpose, as it is able to capture asymmetric high-order proximity in directed networks. Specifically, in a directed network[2], if there is a directed link from vertex $v_i$ to vertex $v_j$ and from vertex $v_j$ to vertex $v_k$, it is more likely to have a link from $v_i$ to $v_k$, but not from $v_k$ to $v_i$. In order to preserve such asymmetric transitivity, HOPE learns two vertex embedding vectors $U^s, U^t \in \mathbf{R}^{|V| \times d}$, which is called source and target embedding vectors, respectively. After constructing the high-order proximity matrix $S$ from $4s$ proximity measures, i.e., Katz Index, Rooted PageRank, Common Neighbors and AdamicAdar. HOPE finally learns vertex embeddings by solving the a matrix factorization problem:

$$\min_{U_s, U_t} \|S - U^s U^{t^T}\|_F^2 \tag{1}$$

**UMAP Dimensionality Reduction.** While HOPE (as well as any other node embedding methods) could generate vectorized embeddings for configurations, these embeddings typically are still high-dimensional and not compatible for 2D visualization. To cope with this, we further apply UMAP[3] to project the HOPE embeddings into a 2D space, which is based on three assumptions, and here we provide brief discussion on whether they hold for our case:

- **Data are uniformly distributed on Riemannian manifold.** As the distribution of HOPE embeddings essentially depends on the connectivity pattern of the graph, for HP loss landscapes, we can expect this assumption to hold. This is because the number of neighbors for each node in the graph would largely stay the same based on our neighborhood definition. Therefore, there will be no significance variation of *density* across the graph, and the HOPE embeddings in turn should approximately have a uniform distribution.

---

[1]We use the implementation in Karateclub package.

[2]Most manipulations and analyses of HP loss landscapes as complex networks are grounded on NetworkX and Pandas package.

[3]We use the implementation in UMAP package.

---

**Algorithm 1** Best-Improvement Local Search

---

**Require:** A starting configuration $\mathbf{c}$; A neighborhood function $\mathcal{N}_d$; A fitness function $f$
1: **while** $\mathbf{c}$ is not a local optimum **do**
2:     $\mathbf{c}'_{\text{best}} = \arg\min_{\mathbf{c}' \in \mathcal{N}_1(\mathbf{c})} f(\mathbf{c}')$
3:     **if** $f(\mathbf{c}'_{\text{best}}) < f(\mathbf{c})$ **then**
4:         $\mathbf{c} \leftarrow \mathbf{c}'_{\text{best}}$
5:     **end if**
6: **end while**
7: **return** $\mathbf{c}$

---

---

**Algorithm 2** Identifying Local Optima and Their Basins

---

**Require:** The set of all configurations $\mathcal{C}$
1: $\mathcal{V} \leftarrow \emptyset$
2: $\mathcal{B} \leftarrow \emptyset$
3: **for all** $\mathbf{c} \in \mathcal{C}$ **do**
4:     $\mathbf{c}^\ell \leftarrow \text{LOCALSEARCH}(\mathbf{c})$
5:     $\mathcal{B}[\mathbf{c}^\ell] \leftarrow \mathcal{B}[\mathbf{c}^\ell] \cup \{\mathbf{c}\}$
6: **end for**
7: **return** $\mathcal{V}, \mathcal{B}$

---

- **The Riemannian metric is locally constant.** The distances between HOPE embeddings correlate directly with the local connectivity patterns within the network. Given the stability in neighborhood structures as we discussed above, it is reasonable to presume that the local distance metrics would remain approximately constant in small regions.

- **The manifold is locally connected.** HOPE embeddings can actually preserve more than just the local structure between configurations, as it is able to capture high-order proximities in the graph. We thereby expect this assumption will also hold.

**Remark on Algorithm Choices.** In principle, other node embedding and dimensionality reduction methods can be applied to serve our purpose. Our specific choice on HOPE is mainly because of its scalability to large-scale networks and its ability to preserve both local and global structure of the landscape. As for dimensionality reduction, Draganov et al. (2023) has made detailed theoretical comparisons between UMAP and $t$-SNE, and shows that only the normalization significantly impacts their outputs. This thens implies that a majority of the algorithmic differences can be toggled without affecting the embeddings. We choose UMAP here for its better scalability. At the end, the quality of the visualization using our method will continuously grow with the new state-of-the-art in graph representation learning and dimensionality reduction.

**Remark on UMAP Hyperparameters.** Since HP loss landscapes can vary a lot in terms of dimensionality and total number of configurations, there is no 'one-size-fits-all' setup for HPs of UMAP. However, we still provide some general guidelines for HP tuning, which mainly focus on two most important HPs, namely `n_neighbors` and `min_dist`. Specifically, `n_neighbors` controls the balance between local and global structure in the embedding. In general, we found that it is better to set `n_neighbors` to a value that is larger than the average number of neighbors for each node. For `min_dist`, which specifies the minimum distance between points in the low-dimensional space, we generally recommend values larger than $0.5$. The reasoning here is that we want the points to be more spread out in the low-dimensional space, and thus prevent the points to be densely distributed in local regions. However, a `min_dist` that is too large can also cause problems, as different parts of the landscape can get intertwined with each other.

## B.2 LANDSCAPE ANALYSIS METRICS

**Assortativity coefficient.** The assortativity coefficient of a network assesses the degree to which nodes tend to be connected to other nodes that are similar w.r.t. some attributes. For example, in a social network, this would mean that people tend to be friends with other people who are similar to

themselves in terms of education level, income, race[4], etc. For HP loss landscape, the performance assortativity measures the extent to which HP configurations with similar performance are more likely to be neighbors to each other. Formallly, given a HP loss landscape as a directed graph, and the model loss $\mathcal{L}$ takes values $[L_1, L_2, \dots]$, the $\mathcal{L}$-assortativity evaluates the Pearson correlation of the measured loss between pairs of linked configurations and is measured as (Newman, 2003):

$$\mathcal{L}\text{-ast} = \frac{\sum_i e_{ii} - \sum_i a_i b_i}{1 - \sum_i a_i b_i} \tag{2}$$

where $e_{ij}$ is called mixing matrix entry, which represents the fraction of total edges in the network (i.e., landscape) which connects configurations having performance $\mathcal{L}(\boldsymbol{\lambda}) = L_i$ to configurations having attribute $\mathcal{L}(\boldsymbol{\lambda}) = L_j$. In directed networks like our case, this can be asymmetric, i.e., $e_{ij} \neq e_{ji}$. In addition, $a_i = \sum_j e_{ij}$ is the portion of edges $(\boldsymbol{\lambda}_u, \boldsymbol{\lambda}_v)$ such that $\mathcal{L}(\boldsymbol{\lambda}_u) = L_i$ and $b_i = \sum_j e_{ji}$ is the portion of edges $(\boldsymbol{\lambda}_v, \boldsymbol{\lambda}_u)$ such that $\mathcal{L}(\boldsymbol{\lambda}_v) = L_i$. A high $\mathcal{L}$-ast would imply that configurations with similar performance have strong tendancy to be connected and form local clusters.

**Landscape Neutrality.** It is often in genetics that a mutation in a single position of a DNA sequence will only lead to negligible change in the expression. Such phenomenon is known as landscape *neutrality* at a macro level, and for each sequence, this can be quantitatively measured by the *mutational robustness* (Payne & Wagner, 2019), which is the probability for such non-effective mutation to happen among all its possible mutants. Similar ideas are also applicable to HPO, where altering certain HPs may only result in subtle performance shifts. In particular, we define two neighboring configurations to be *neutral* if their respective performances differ less than a small fraction $\epsilon$. We choose $\epsilon = 0.1\%$ in this paper, since changes below this threshold would make almost no practical meaning. We then define the *neutral ratio*, denoted as $\nu(\boldsymbol{\lambda})$, of a configuration $\boldsymbol{\lambda}$ as the portion of its neutral neighbors in its neighborhood. The avarage neutrality of the whole landscape is then defined as:

$$\bar{\nu} = \mathbb{E}[\nu(\boldsymbol{\lambda})] = \frac{1}{|\boldsymbol{\Lambda}|} \sum_{\boldsymbol{\lambda} \in \boldsymbol{\Lambda}} \nu(\boldsymbol{\lambda}) \tag{3}$$

**Neutrality Distance Correlation.** While neutrality generally characterizes the expected probability for neutral moves to occur in the whole landscape, it can actually vary across regions. In particular, it is important to investigate whether it is more likely to encounter neutral moves when approaching the global optimum, as in practice, we often find a diminishing gain when tuning towards the best-possible configuration. We quantitatively assess this using the neutrality distance correlation (NDC), which measures the Pearson correlation coefficient between the neutrality of a configuration $\boldsymbol{\lambda}$ and its distance to the global optimum, $\mathrm{d}(\boldsymbol{\lambda}, \boldsymbol{\lambda}^*)$ (equation (4)). We need to note that result derived from this can be potentially affected by the choice of $\epsilon$ for neutrality. To cope with this, we also develop an alternative method for assessing NDC, which is directly based on raw loss differences between configurations. Specifically, for each adaptive walk in the landscape using best-improvement local search (Algorithm 1) that can eventually approach $\boldsymbol{\lambda}^*$, we measure the Pearson correlation coefficient between $\Delta\mathcal{L}$ for each pair of consecutive configurations $(\boldsymbol{\lambda}_i, \boldsymbol{\lambda}_{i-1})$ $(i \geq 2)$, and $\mathrm{d}(\boldsymbol{\lambda}_i, \boldsymbol{\lambda}^*)$. We calculate NDC as the average across all such walks. However, in this paper, to keep the consistency with the neutrality metric, we report the results based on the first method.

$$\text{NDC} = \rho_p[\nu(\boldsymbol{\lambda}), \mathrm{d}(\boldsymbol{\lambda}, \boldsymbol{\lambda}^*)] \tag{4}$$

**Number of Local Optima.** A configuration $\boldsymbol{\lambda}$ is said to be a *local optimum* if its performance is superior to any other configuration in its neighborhood, i.e., $\forall \boldsymbol{\lambda} \in \mathcal{N}(\boldsymbol{\lambda}^\ell)$, we have $\mathcal{L}(\boldsymbol{\lambda}^\ell) < \mathcal{L}(\boldsymbol{\lambda})$. For a *unimodal* landscape, there is only a global optimum configuration $\boldsymbol{\lambda}^*$. In constrast, multimodal landscapes have various local optima with sub-optimal performance, which can pose challenges to the optimization.

**Size of Basin of Attraction.** While a multimodal landscape can be difficult to optimize due to the pressence of various local optima, not all of them are equal in terms of the capability of trapping a solver. For a 2D minimization scenario, this can be envisioned by the fact that each local optimum is located at the bottom of a 'basin' in the landscape surface. Configurations in each basin would eventually fall into the corresponding basin bottom (i.e., local optimum) when applying a simplest hill-climbing local search (Algorithm 1). The effort needed to escape from such a basin is direclty

---

[4]As one can image, the exact definition of assortativity can depend on whether the target attribute is categorical (i.e., unordered) or numerical (i.e., ordered). Here we focus on the latter as model loss is real-valued.

---

**Algorithm 3** Constructing Local Optima Network

---

**Require:** The set of local optima $\mathcal{V}$; The basin of attraction of each local optimum $\mathcal{B}_{\mathbf{c}^\ell}$; The set of all configurations $\mathcal{C}$; A neighborhood function $\mathcal{N}_d(\mathbf{c})$

1: $\mathcal{E} \leftarrow \emptyset$
2: $\mathcal{W} \leftarrow \emptyset$
3: **for all** $\mathbf{c}^\ell \in \mathcal{V}$ **do**
4:     **for all** $\mathbf{c}^{\ell\prime} \in \mathcal{N}_2(\mathbf{c}^\ell)$ **do**
5:         $\mathbf{c}^\ell_{\text{new}} \leftarrow \text{LOCALSEARCH}(\mathbf{c}^{\ell\prime})$
6:         **if** $f(\mathbf{c}^\ell_{\text{new}}) < f(\mathbf{c}^\ell)$ **then**
7:             **if** edge $(\mathbf{c}^\ell, \mathbf{c}^\ell_{\text{new}})$ not in $\mathcal{E}$ **then**
8:                 $\mathcal{E} \leftarrow \mathcal{E} \cup \{(\mathbf{c}^\ell, \mathbf{c}^\ell_{\text{new}})\}$
9:                 $\mathcal{W}[(\mathbf{c}^\ell, \mathbf{c}^\ell_{\text{new}})] \leftarrow 1$
10:             **else**
11:                 $\mathcal{W}[(\mathbf{c}^\ell, \mathbf{c}^\ell_{\text{new}})] \leftarrow \mathcal{W}[(\mathbf{c}^\ell, \mathbf{c}^\ell_{\text{new}})] + 1$
12:             **end if**
13:         **end if**
14:     **end for**
15: **end for**
16: **return** $\mathcal{G} = (\mathcal{V}, \mathcal{E}, \mathcal{W})$

---

related to its sizes (e.g., depth and radius in a 2D space). A local optimum with small basin size is very unlikely to cause significant obstacles to optimization, whereas the opposite is true for those featuring a dominated size of basin that is even larger than the global optimum. Formally, we define the *basin of attraction* $\mathcal{B}$ of a local optimum $\boldsymbol{\lambda}^\ell$ to be the set of all configurations from which local search converges to $\boldsymbol{\lambda}^\ell$, i.e., $\mathcal{B} = \{\boldsymbol{\lambda} \in \boldsymbol{\Lambda} \mid \text{LocalSearch}(\boldsymbol{\lambda}) \to \boldsymbol{\lambda}^\ell\}$ (Algorithm 2). The size of $\mathcal{B}$, denoted as $s_\mathcal{B}$, is defined as the cardinality of the basin set as $|\mathcal{B}|$. In this study, we report the mean basin size $\bar{s}_\mathcal{B}$ of all local optima (except the global optimum) in the landscape.

**Autocorrelation.** A common metric for characterizing the smoothness of a landscape is the auto-correlation $\rho_a$ on a series of performance values $\mathcal{L}$. These values are extracted for configurations in a random walk $\text{RW} = \{\boldsymbol{\lambda}_0, \boldsymbol{\lambda}_1, \ldots, \boldsymbol{\lambda}_n\}$ in the search space $\boldsymbol{\Lambda}$. Formally:

$$\rho_a(k) = \frac{\mathbb{E}[(\mathcal{L}(\boldsymbol{\lambda}_i) - \bar{\mathcal{L}})(\mathcal{L}(\boldsymbol{\lambda}_{i+k}) - \bar{\mathcal{L}})]}{\mathbb{V}(\mathcal{L}(\boldsymbol{\lambda}_i))}, \forall \boldsymbol{\lambda}_i \in \boldsymbol{\Lambda} \tag{5}$$

Here, $k$ represents a lag or a step difference in the indices of configurations, and in our case we consider $k = 1$ since each step in our search grids have been specifically designed to mimic the tunning strategy commonly used by human experts. For each landscape, we conduct 100 random walks of length 100 and average $\rho_a$ across all measurements to mitigate the effects of randomness.

### B.3 LOCAL OPTIMA NETWORK

Beyond the number of local optima and their basin sizes, a further aspect to investigate is the *connectivity* pattern between them. For example, an important question that we may concern is *whether we can escape from a given local optimum to the global optimum?*, if yes, then *what is the chance of this?* Local optima networks (LON) (Ochoa et al., 2008; Vérel et al., 2011), which are rooted in the study of energy landscapes in chemical physics (Stillinger, 1995), address these questions by constructing a subspace of the original fitness landscape where the nodes indicate local optima, and edges represent possible transitions between them. In particular, an improving edge can be traced from local optimum configuration $\boldsymbol{\lambda}_i^\ell$ to $\boldsymbol{\lambda}_j^\ell$ if configurations in $\mathcal{B}(\boldsymbol{\lambda}_i^\ell)$ can escape to $\boldsymbol{\lambda}_j^\ell$ by applying a 2-bit perturbation followed by local search. The edge weights $w_{i,j}$ indicate the total probability for such transitions to happen (see Algorithm 3) between two local optima. By conducting network mining on LONs (e.g., Huang & Li (2023)), we can get further insights into the distribution and connection between local optima as well as how they are potentially linked with the global optimum. However, while we believe LON could be an effective tool for analyzing complex landscapes and we incorporate this as part of our landscape analysis tool, we do not present such analyses in most scenarios in this paper since their HP loss landscapes only possess a few local optima (if not none).

### B.4 LANDSCAPE SIMILARITY METRICS

**Spearman Correlation.** It is a non-parametric measure of rank correlation which assesses how well the relationship between configuration performances in two landscapes can be described using a monotonic function. It is defined as the Pearson correlation coefficient between the performance ranks configurations in two landscapes.

**Shake-up Metric.** It originates from the Kaggle competition community and is designed to assess the rank changes between the public and private leaderboards of a competition (Trotman, 2019). To be specific, it quantifies the average movement of rankings from public board to the private board. For HP loss landscapes, this metric can indicate the expected rank shifts for a configuration when evaluating it on two different scenarios (e.g., change the dataset).

$$\text{Shake-up} = \mathbb{E}\left[\frac{|R(\mathcal{L}_1(\boldsymbol{\lambda})) - R(\mathcal{L}_2(\boldsymbol{\lambda}))|}{|\boldsymbol{\Lambda}|}\right] = \frac{1}{|\boldsymbol{\Lambda}|}\sum_{\boldsymbol{\lambda}\in\boldsymbol{\Lambda}}\frac{|R(\mathcal{L}_1(\boldsymbol{\lambda})) - R(\mathcal{L}_2(\boldsymbol{\lambda}))|}{|\boldsymbol{\Lambda}|} \tag{6}$$

$\gamma$**-set Similarity.** It is proposed in Watanabe et al. (2023a) to assess the similarity of two tasks using the ratio of their most prominent configurations. More specifically, for two HP loss landscapes, their $\gamma$-set similarity is defined as the ratio of the intersection of the top-$\gamma$ regions to the union of them. In this paper, we consider $\gamma = 10\%$.

## C ANALYSIS ON NAS LOSS LANDSCAPES

### C.1 NAS BENCHMARKS

**NAS-Bench-101.** It represents the first effort towards benchmarking NAS research and thus fostering reproducibility in the community. It evaluates a CNN architecture with 3 stacked blocks, where a down-sampling is added between two consecutive blocks; A $3 \times 3$ convolution is used before the main blocks, and the outputs of the main blocks are fed to an average pooling and fully connected layer. NAS-Bench-101 considers a *cell*-based search space which enumerates all possible configurations for each block. More specifically, the search space is formulated as a directed acylic graph (DAG) with 7 nodes and a maximum of 9 edges. Here, each node can represent one of the following operations: $a$): $1 \times 1$ convolution, $b$): $3 \times 3$ convolution, and $c$): max pooling. After removing all isomorphic cells, this search space results in 423k unique configurations. NAS-Bench-101 evaluates each of them on the CIFAR-10 dataset and records meta-data at the $\{4, 12, 36, 108\}^{\text{th}}$ epoch.

**NAS-Bench-201.** It features a different skeleton compared to NAS-Bench-101, in which a residual block is applied to connect 3 cells. Each cell here is a DAG with 4 nodes and 6 edges. Morever, here, operations are represented by edges, which have 5 types: $a$): zeroize (do nothing), $b$): $1 \times 1$ convolution, $c$): $3 \times 3$ convolution, $d$): $3 \times 3$ average pooling, $e$): skip connection. The benchmark thus contains $5^6 = 15,625$ unique model architectures, with each evaluated on 3 different datasets: $i$): CIFAR-10, $ii$): CIFAR-100, $iii$): ImageNet-16-120 using 200 epochs.

### C.2 NAS LANDSCAPE CONSTRUCTION

Despite the cell-based search spaces of NAS benchmarks are very different from the HP ones considered in this paper, our landscape construction routine could be easily transfered to NAS by redefining the neighborhood structure. This then demands proper encoding of the NAS configurations and the definition of a suitable distance function.

**NAS-Bench-101 Neighborhood.** In the original paper, a cell-encoding method based on adjacency matrices is introduced to encode configurations, which comprises two components. First, a $7 \times 7$ upper-triangular binary matrix is used to indicate whether an edge exist between two nodes and thus determine the connectivity pattern of nodes. Next, the functionality of a configuration also depends on which operation is performed at each node, and this could be encoded via a vector of length 5 (the input and output nodes are the same across archiectures and are omitted, non-existent nodes are represented by `NaN`). Therefore, a configuration could be specified using an adjacency matrice and a node vector. We define a neighbor of a given configuration to be the one with 1-edit distance from it (e.g., either adding or deleting one edge, or change the operation of one node), as in the original paper. Note that not all such configurations are valid in the benchmark, as they can be isomorphic to others. The benchmark API provides built-in function to check for this.

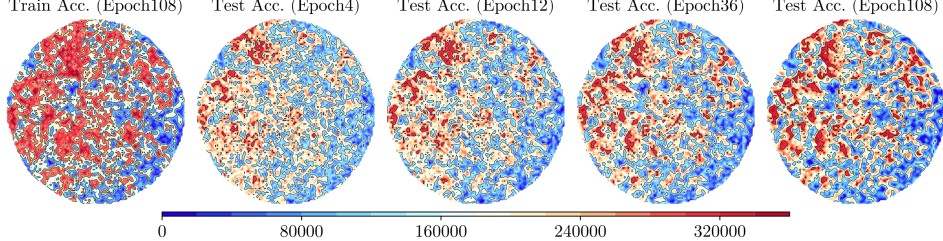

Figure 8: Visualization of NAS-Bench-101 landscape using our proposed method. Here, we present the training accuracy landscapes as well as test accuracy landscapes at different recorded epochs. Color indicates rank of test accuracy, and higher values are better.

**NAS-Bench-201 Neighborhood.** The encoding of configurations in this benchmark is much more straight forward. Specifically, we encode each configuration using a 6-bit vector, where each bit specifies the operation taken at each edge, which can take 5 categorical values. We base our neighborhood definition here on the 1-edit distance as well, in which a 1-bit mutant of a configuration is the one with the operation in only one edge altered.

## C.3    RESULTS ON NAS LOSS LANDSCAPES

While we have conducted all analyses on both benchmarks, it would be too tedious to lay all the information here. Instead, we use NAS-Bench-101 to discuss general NAS landscape characteristics and multi-fidelity, and present the comparisons between NAS landscapes across datasets using NAS-Bench-201. Before we present our results, we also note that while the majority of configurations (359k out of 423k) in NAS-Bench-101 search space come with 7 nodes (i.e., with 5 intermediate operations), the rest of them have less nodes. Here we mainly focus on configuragtions with 7 nodes, since accounting all the configurations would result in many independent components in the landscape. We do not expect this to significantly affect our results.

**NAS Landscape Visualization.** We first visulize the NAS-Bench-101 landscape using our proposed landscape visualization method in Section 2, as shown in Figure 8. Specifically, we plot the training accuracy landscape along with test accuracy landscapes trained at 4 different number of epochs. It is clear to see from the plot that the landscape is far from unimodal, with many local optima. However, configurations still tend to form local clusters, though the relative size of each plateau seems to be much smaller than we see for HP loss landscapes. Considering that the search space here contains nearly 30 times more confiurations than the HP space we used in the main text, each cluster may contain thousands of configurations, inside which the landscape could still be sufficiently smooth.

**NAS Landscape Metrics.** Here we report several landscape metrics for the NAS-Bench-101 landscape with respect to test accuracy at the 108-*th* epoch:

- **Autocorrelation.** We obtained a correlation coefficient of 0.6031 on the landscape. This confirms our hypothesis above that the landscape is still sufficiently smooth and highly navigable, despite we observed more complex patterns in the visualizations.

- **Clustering.** The *accuracy*-assrotativity is 0.6485, which indicates a good level of local clustering of configurations with similar performance levels. This further confirms that the landscape is locally smooth.

- **Neutrality.** Our neutrality measure, however, only yields a value of 0.075, and suggests that most 1-bit changes in a configuration would result in performance shift $> 0.1\%$.

- **NDC.** Despite the overall neutrality of the NAS landscape is low, we still observe a high NDC value of 0.7194. This implies a strong plateau trend near the optimum, where optimizers would pay considerable more effort to gain marginal performance improvement.

- **Number of Local Optima.** One of the most distinguishable property of NAS-Bench-101 is a large collection of local optima in the landscape. In fact, we found 5,908 local optima in total (out of the 359k configurations), which could probably make the landscape far more

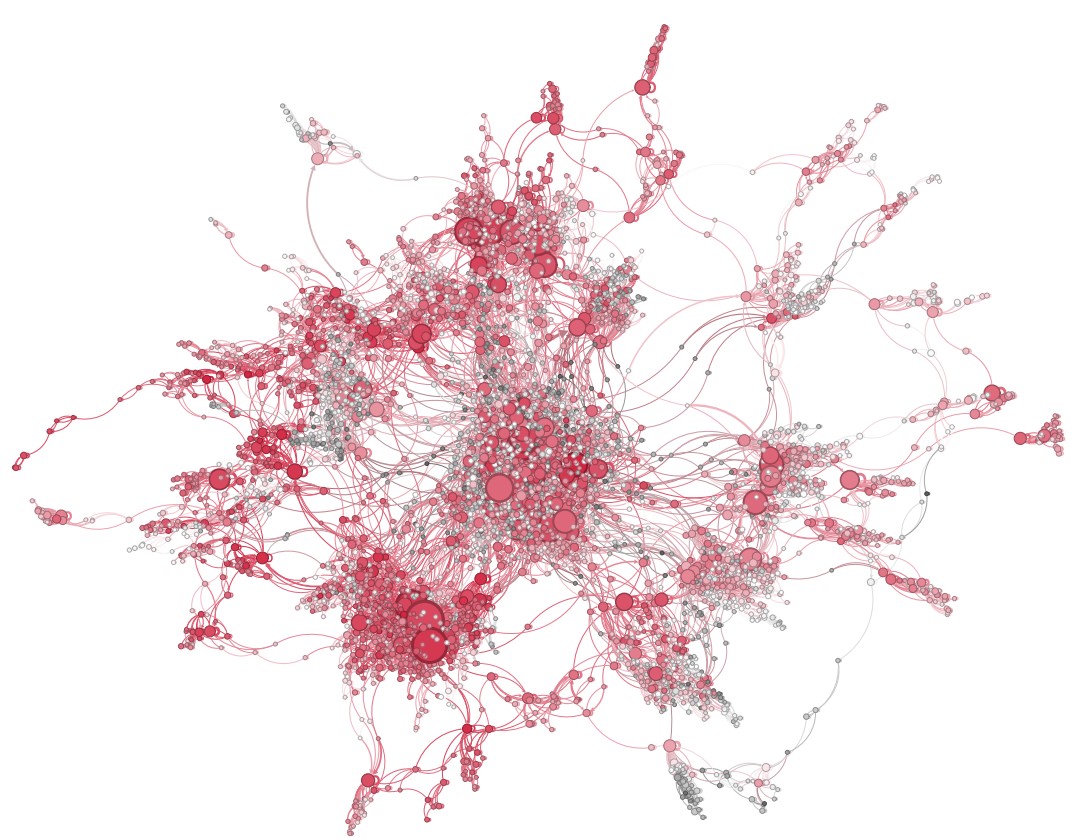

Figure 9: Visualization of the local optima network of NAS-Bench-101 test landscapes at the 108-*th* epoch, containing $5,908$ nodes. Radius of each node (i.e., local optimum) indicates size of the corresponding basin of attraction. The color indicates test accuracy, where warmer color is better. Edges indicate transition probabilities between local optima, where thicker, warmer edges imply the corresponding transition if more likely to happen. Edge directions indicate the improving direction (i.e., pointing to the fitter configuration).

difficult to optimization. We will discuss more about them and their basins in the LON part.

**NAS Landscape Across Fidelities.** From Figure 8, we could see that in general the test landscapes with lower fidelities resemble the one trained with 108 epoch. Quantitatively, the Spearman correlation between the test accuracy landscapes at 108-*th* epoch and 32-*th* epoch is 0.904, with a Shake-up metric of 9.74%. These suggest a good general correlation between these landscapes, although there are 4 times difference in their budget. When zooming into the 10% region, the $\gamma$-set similarity is 0.64, which is also farily good (the intesection ratio between top-10% regions is 78%). When we further decrease the budge by 4 times, the Spearman correlation and Shake-up metric obtained between the 12-*th* and 108-*th* epoch are 0.657 and 18.4% respectively, while the $\gamma$-set similarity is 0.317. Finally, with only 4 epochs of training, the above metrics further change to 0.504, 22.4% and 0.164 respectively. In general, the landscapes are still moderately correlated, but the detailed pattern can be largely distorted.

**Local Optima Network Analysis.**

Figure 9 shows the local optima network of NAS-Bench-101 test landscapes at the 108-*th* epoch. It could be obviously seen that there are clear community (clustering) structure in the network, where local optima with large basin of attractions tend to locate at the center of each cluster, which usually feature a promising performance. To be specific, from the left plots in Figure 11, we could

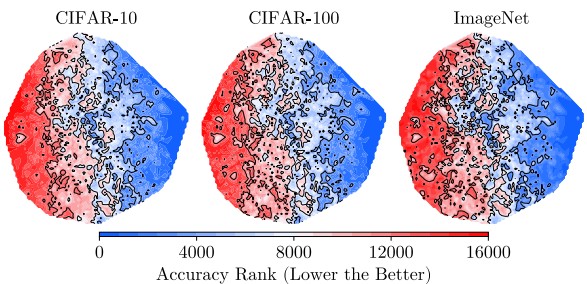

Figure 10: Visualization of NAS-Bench-201 landscape using our proposed method across datasets.

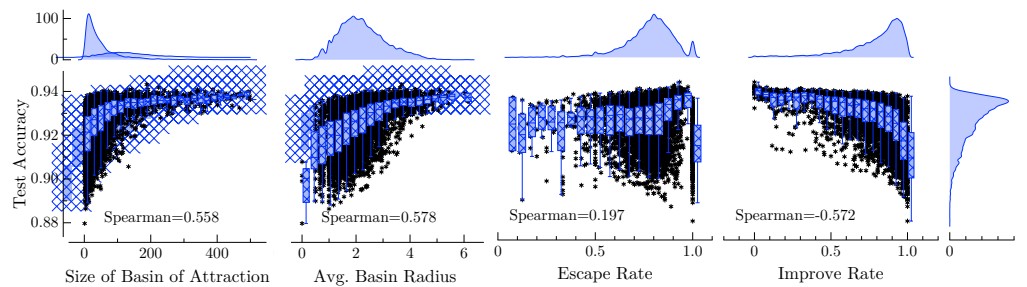

Figure 11: Distribution of local optima test accuracy versus LON metrics.

clearly observe that the size and radius[5] of the basin of attraction is positively correlated with the performance of local optima (Spearman correlation $> 0.55$). More importantly, as suggested by the cumulative distribution of basin size shown in Figure 12, the highly-fit local optima have a dominant size of basin of attractions in the landscape. For example, the dashed line in Figure 12 indicates that the cumulative sum of basin sizes of those local optima with $acc > 94.3\%$ take $50\%$ of the total basin size (which is the number of total configurations in the landscape). Since being in the basin of a local optimum would imply that local search will eventually converge it, our result is to say that if we start local search from a random configuration in the landscape, there is $50\%$ chance that we would end up in a local optimum with $acc > 94.3\%$. This is a very promising result, since we find such a level of accuracy is already better than $98.58\%$ of total configurations in the whole search space! It is also superior than $76.84\%$ of other local optima in the landscape.

A natural question that follows is the efforts that we need to reach such local optima. Statistically, we find that on average, it would take 3.04 local search steps to reach a local optimum, while the mean of the longest walk length in each basin is 6.46 steps. This is not a huge effort, since after taking such steps, as we discussed above, we have a good chance to fall into local optimum with $acc > 94.3\%$. However, we note that this is also not that trivial, since each 'step' here means we exhaustively search for all the neighbors of a configuration, and then select the best one to move on. Such local search technique is called a *best-improvement* local search. In contrast, there is also *first-improvement* local search, in which we take the first configuration in the neighborhood that is fitter than the current one without considering other neighbors. This in general woul require more steps to reach a local optimum, but with fewer model evaluations at each step. While local search is definitely sub-

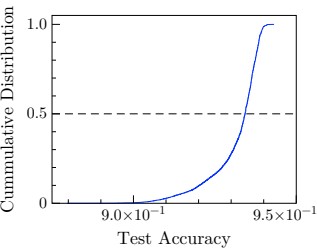

Figure 12: Cumulative distribution of basin size versus local optima performance.

optimal compared to advanced global search strategies, finding that even such a simple technique

---

[5]Here, we define the *radius* of a basin of attraction as the expected number of local search steps needed to reach the corresponding local optimum. Since in practice, we observe that this value is often highly correlated with the size of basin of attraction, and correlate with performance in a similar manner, we only report basin size in the main text.

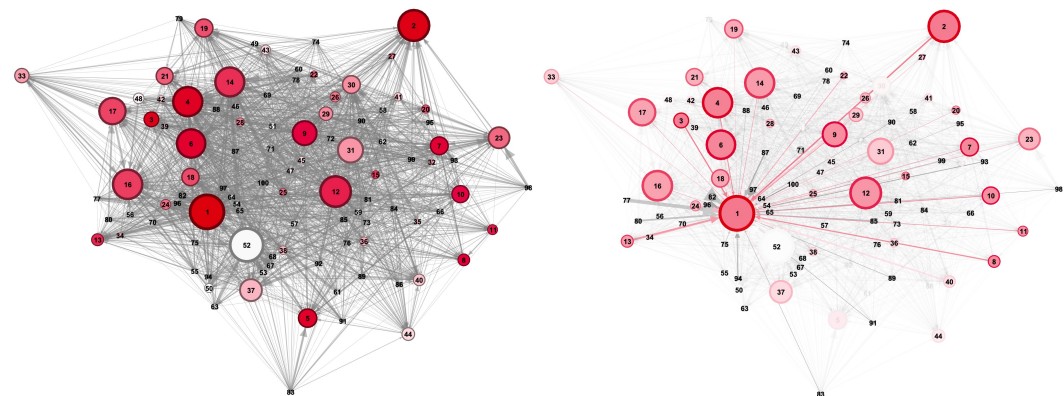

Figure 13: Local optima network for the 100 local optima in the FCNet landscape on parkinsons telemonitoring dataset. The left plot shows the full-view of the network, where node size represents the size of the corresponding basin of attraction. Node color indicates the performance of the model, which is labeled as rank values for each node, and the global optimum has a rank of 1. The right plot shows the neighborhoods of the global optimum, while other nodes are omitted.

could somehow lead to a guaranteed promising result would imply that the NAS landscape is also benign.

Beyond local search, we then continue to consider if we do fall in a local optima whose fitness is not that satisfactory, then *what is the chance that we can somehow escape from it?*. The two plots at the right panel of Figure 11 show the distribution of *escape rate* and *improve rate* correlate with test accuracy. Here, the escape rate is the chance that, a configuration in the basin of a local optimum $c^\ell$, after applying a 2-bit perturbation, would converge to a different local optimum $c^\ell_{new}$. On the other hand, the improve rate further restricts that the new local optimum should have better performance compared to the current one. From Figure 11, we could clearly see observe the majority of the local optima feature a escape rate of larger than 50%, which suggests that most local optima are not that difficult to escape from. For improve rate, we observe a good correlation with test accuracy, where it is more easy to find an improving move for a poorly-performed local optimum. Unfortunately, for those that already have promising performance, there is only little chance to transit to a better basin using 2-bit perturbation.

### C.4 Results on NAS-Bench-201

We visualize the loss landscapes of NAS-Bench-201 on the 3 datasets, namely CIFAR-10/100 and ImageNet, in Figure 10. In general, we see that results on these 3 tasks reveal strong consistency to each other (Spearman correlation > 0.95), which conforms with our findings on HP loss landscapes.

## D Details About FCNet HP Loss Landscapes

Table 3: Landscape Features of FCNet on Different Datasets

| Dataset | Autocorr. | Assortativity | Neutrality | NDC | #LO | Mean Basin Size |
|---|---|---|---|---|---|---|
| Parkinsons Tele. | 0.4683 | 0.6232 | 0.4317 | 0.6368 | 100 | 589 |
| Protein Struc. | 0.4896 | 0.5789 | 0.4746 | 0.7411 | 128 | 385 |
| Naval Prop. | 0.5720 | 0.5298 | 0.4091 | 0.6433 | 347 | 159 |
| Slice Local. | 0.4887 | 0.6134 | 0.4427 | 0.8216 | 24 | 2372 |

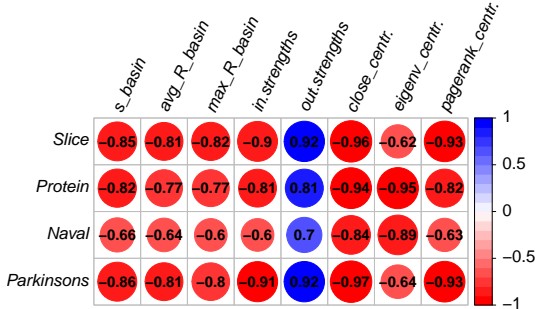

Figure 14: Spearman correlation between LON features ($i$): mean basin size, $ii$): mean basin radius, $iii$): maximum basin radius, $iv$): in-coming strengths, $v$): out-going strengths, $vi$): closeness centrality, $vii$): eigenvector centrality, $viii$): pagerank centrality) and local optima performance for `FCNet` across datasets.

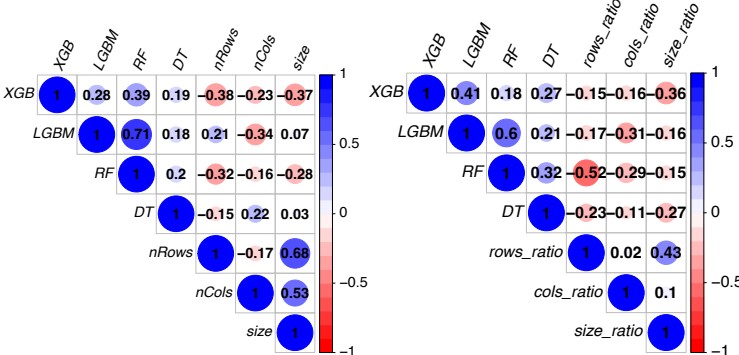

Figure 15: Spearman correlation between landscape similarity and dataset characteristics. Left: similarity between training and test loss landscapes for different models versus dataset characteristics. Right: similarity between landscape on different dataset for different models versus ratio of dataset characteristics. Here, dataset *size* is defined as the product of the number of instances and features.

### D.1 LANDSCAPE FEATURES

From the visualization of FCNet landscapes shown in Figure 2, we can conclude that they generally follow the same characteristics as we discussed in Section 4.1. Most of such observations are supported by FLA metrics reported in Table 3. However, we note that all four landscapes seem to reveal multimodal structures, as there are dozens to hundreds of local optima with relative large basins (up to a mean size of $2,372$). Similar observations have been made by Pushak & Hoos, who speculated that the reasons could be `FCNet` scenarios fall into the over-parameterized regime, which is different from the other scenarios. While we also agree with this reasoning, we seek to conduct further analysis on the local optima using the local optima network (LON) (Appendix B.3). Figure 13 (left) shows the LON of `FCNet` on the parkinsons telemonitoring dataset. It can be seen that in general, local optima with better performance tend to feature a large basin of attraction and high in-coming strenghts (i.e., in LON context, the total probability for other local optima to reach it). This implies that, despite the pressence of many other local optima, the global optima still plays a pivotal role in the connectivity of the network (see Figure 13, right), which is not the worst case that one may encounter (e.g., a global optimum with tiny basin locates at a secluded region that the optimizer can hardly find.)

## E LANDSCAPE SIMILARITY & DATASETS CHARACTERISTICS

While in general, we find that HP loss landscapes studied in this paper share various common characteristics, the detailed topography can vary with dataset that the model is trained on. How charac-

teristics of datasets could affect the landscape is an interesting problem to explore in more detail. In addition, we also hypothesis that the generalization gap of a model can also depend on the dataset, in addition to its HP setting. To investigate these, we conduct additional experiments on the 57 tabular datasets to analyze the relationship between landscape similarity and dataset characteristics.

We first find in the left plot of Figure 15 that the similarity between test and training loss landscapes is positively correlated across models. It implies that on certain datasets, all the 4 models are more prone to overfit, whereas the opposite could be true for other datasets. This then verifies our hypothesis that overfitting not only depends on the model HPs, but also the dataset itself. To further explore which properties of the dataset could potentially contribute to this, we investigated the correlation between train-test landscape similarity and the number of instances & features of each dataset, and their product. The results (also in Figure 15 (left)) indicate that the degree of overfitting is correlated with all 3 dataset size measures. In particular, for most models, it would be more likely to encounter overfitting on larger datasets.

We then proceed to investigate whether correlations between HP loss landscapes induced on different datasets are related to the relative size of the datasets. From Figure 15 (right), it is clear to see that pairwise landscape similarities are obviously correlated across models, implying that all the models are likely to induce very different (or the opposite) landscapes on certain pairs of datasets. We could also see that these pairwise similarities are again correlated with the differences in dataset sizes. In particular, for datasets that have very different sizes, the resulting HP loss landscapes also tend to be different from each other.

However, we note that the correlations reported in both scenarios are somewhat weak, and the reasons could be two folds. First, the choice of datasets is only a partial factor contributing to landscape similarity, many other factors like model HPs, training settings can also important roles here. Second, the dataset meta-features we used here are rather naive, and there could be more comprehensive features for characterizing dataset properties, e.g., Feurer et al. (2015b) had used a collection of 46 features to assess dataset similarity. However, this is beyond the scope of this work and we leave it to future works.

## F  DETAILS OF THE EXPERIMENTAL SETUP

### F.1  SEARCH SPACES

In this subsection, we elaborate the principles that we follow in designing the search spaces for each model. We also provide the detailed hyperparameter grid space for them in Table 4 (`XGBoost`), Table 5 (`DT`), Table 6 (`RF`), Table 7 (`RF`), Table 8 (`CNN`) and Table 9 (`FCNet`). The top-level principle we follow is to include commonly used HPs and exclude unimportant ones, while keeping a good balance between search space coverage and computational cost. We first determine the list of HPs to be considered, and to this end:

- We surveyed commonly used HPs in the practice (e.g., van Rijn & Hutter identified important HPs of several models using large-scale meta-data from OpenML) and HPO literature (e.g., the search spaces used in HPOBench (Eggensperger et al., 2021)). For `CNN`, we also refer to the design of NAS search spaces (Chitty-Venkata et al., 2023).

- We also run preliminary experiments by fixing all except but one HP to their default values, and vary a single HP using a wide range of values. This allows us to estimate the influence of each HP on model performance without conducting large-scale search.

We then combine the knowledge obtained in these precedures to design a rough HP list to be studied for each model. We then proceed to determine the domain and granularity of each HP, for which we bear the following considerations:

- The domain of each HP should be large enough to cover the range of values that are commonly used in practice, while keep a balance with the computational cost. For example, we can not afford to search for `XGBoost` with thousands of base learners.

- Given the limited computational budget, there should be more number of bins for HPs that are more important, e.g., learning rates.

- Given the limited computational budget, we would prefer slightly removing 1 or 2 less important HPs, than having a search grid with many real-valued HPs have only 2 to 3 bins.

Following these principles, our final search spaces generally contain 5 to 8 HPs, with total number of configurations ranging from $6,480$ to $24,200$. We consider these search spaces representative of the real-world practice and thus form a good basis for landscape analysis.

Table 4: `XGBoost` HP Grid Space ($14,960$ Configurations)

| Hyperparameter | Grid Values | Count |
|---|---|---|
| `learning_rate` | $[1e^{-3}, 1e^{-2}, 3e^{-2}, 5e^{-2}, 7e^{-2}, 1e^{-1}, \ldots 5e^{-1}]$ | 11 |
| `subsample` | $[0.2, 0.4, 0.6, 0.8, 1]$ | 5 |
| `max_depth` | $[4, 5, 6, \ldots, 20]$ | 17 |
| `max_bin` | $[256, 512, 1024, 2048]$ | 4 |
| `n_estimators` | $[100, 200, 300, 500]$ | 4 |

Table 5: `DT` HP Grid Space ($24,200$ Configurations)

| Hyperparameter | Grid Values | Count |
|---|---|---|
| `splitter` | $[\texttt{"best"}, \texttt{"random"}]$ | 2 |
| `min_samples_split` | $[2, 4, \ldots, 20]$ | 10 |
| `max_samples_leaf` | $[1, 2, 4, \ldots, 20]$ | 11 |
| `max_depth` | $[5, 10, \ldots, 50, \texttt{None}]$ | 11 |
| `max_features` | $[0.1, 0.2, \ldots, 1.0]$ | 10 |

Table 6: `RF` HP Grid Space ($11,250$ Configurations)

| Hyperparameter | Grid Values | Count |
|---|---|---|
| `min_samples_split` | $[2, 5, 10, 15, 20]$ | 5 |
| `max_samples_leaf` | $[1, 5, 10, 15, 20]$ | 5 |
| `Bootstrap` | $[\texttt{True}, \texttt{False}]$ | 2 |
| `max_features` | $[0.2, 0.4, \ldots, 1.0]$ | 5 |
| `n_estimators` | $[50, 100, 150, 200, 300]$ | 5 |
| `max_depth` | $[10, 15, \ldots, 50, \texttt{None}]$ | 9 |

## F.2 DATASETS

Here we provide basic information regarding the 5 groups of datasets used for this study, including: $i$): numerical regression (Table 10), $ii$): numerical classification (Table 11), $iii$): categorical regression (Table 12), $iv$): categorical classification (Table 11), and $v$): image classifcation (Table 14).

Table 7: `LightGBM` HP Grid Space (13, 440 Configurations)

| Hyperparameter | Grid Values | Count |
|---|---|---|
| learning_rate | $[1e^{-2}, 5e^{-2}, 7e^{-2}, 1e^{-1}, \ldots 5e^{-1},]$ | 10 |
| boosting_type | ["gbdt", "dart"] | 2 |
| max_depth | $[5, \ldots, 30, \text{None}]$ | 16 |
| n_estimators | [50, 75, 100, 150, 200, 250, 300] | 7 |
| num_leaves | [10, 20, 30, 40, 50, 100] | 6 |

Table 8: `CNN` HP Grid Space (6, 480 Configurations)

| Hyperparameter | Grid Values | Count |
|---|---|---|
| act_fn | ["gelu", "relu", "tanh"] | 3 |
| batch_norm | [True, False] | 2 |
| learning_rate | $[1e^{-4}, 5e^{-4}, 1e^{-3}, 3e^{-3}, 5e^{-3}]$ | 5 |
| batch_size | [128, 256] | 10 |
| drop_out | [0.0, 0.1, 0.25, 0.5] | 4 |
| width_linear | [256, 512, 1024] | 3 |
| width_conv | [32, 64, 128] | 3 |
| #conv_block | [2, 4, 6] | 3 |

Table 9: `FCNet` HP Grid Space (62, 208 Configurations)

| Hyperparameter | Grid Values | Count |
|---|---|---|
| initial_LR | $\{5e^{-4}, 1e^{-3}, 5e^{-3}, 1e^{-2}, 1e^{-2}, .1e^{-1}\}$ | 6 |
| batch_size | {8, 16, 32, 64} | 4 |
| LR_schedule | {cosine, fix} | 2 |
| activation_layer_1 | {relu, tanh} | 2 |
| activation_layer_2 | {relu, tanh} | 2 |
| layer_1_Size | {16, 32, 64, 128, 256, 512} | 6 |
| layer_2_Size | {16, 32, 64, 128, 256, 512} | 6 |
| dropout_Layer_1 | {0.0, 0.3, 0.6} | 3 |
| dropout_Layer_2 | {0.0, 0.3, 0.6} | 3 |

Table 10: Numerical regression

| OpenML ID | Dataset Name | #Samples | #Features |
|---|---|---|---|
| 44132 | cpu_act | 8,192 | 21 |
| 44133 | pol | 15,000 | 26 |
| 44134 | elevators | 16,599 | 16 |
| 44136 | wine_quality | 6,497 | 11 |
| 44137 | Ailerons | 13,750 | 33 |
| 45032 | yprop_4_1 | 8,885 | 42 |
| 44138 | houses | 20,640 | 8 |
| 44139 | house_16H | 22,784 | 16 |
| 45034 | delays_zurich_transport | 5,465,575 | 9 |
| 44140 | diamonds | 5,3940 | 6 |
| 44141 | Brazilian_houses | 10,692 | 8 |
| 44142 | Bike_Sharing_Demand | 17,379 | 6 |
| 44143 | nyc-taxi-green-dec-2016 | 581,835 | 9 |
| 44144 | house_sales | 21,613 | 15 |
| 44145 | sulfur | 10,081 | 6 |
| 44146 | medical_charges | 163,065 | 5 |
| 44147 | MiamiHousing2016 | 13,932 | 14 |
| 44148 | superconduct | 21,263 | 79 |

Table 11: Numerical classification

| OpenML ID | Dataset Name | #Samples | #Features |
|---|---|---|---|
| 44120 | electricity | 38,474 | 7 |
| 44121 | covertype | 566,602 | 10 |
| 44122 | pol | 10,082 | 26 |
| 44123 | house_16H | 13,488 | 16 |
| 44125 | MagicTelescope | 13,376 | 10 |
| 44126 | bank-marketing | 10,578 | 7 |
| 45019 | Bioresponse | 3,434 | 419 |
| 44128 | MiniBooNE | 72,998 | 50 |
| 45020 | default-of-credit-card-clients | 13,272 | 20 |
| 44129 | Higgs | 940,160 | 24 |
| 44130 | eye_movements | 7,608 | 20 |
| 45022 | Diabetes130US | 71,090 | 7 |
| 45021 | jannis | 57,580 | 54 |
| 45089 | credit | 16,714 | 10 |
| 45028 | california | 20,634 | 8 |

Table 12: Categorical regression

| OpenML ID | Dataset Name | #Samples | #Features |
|---|---|---|---|
| 45041 | topo_2_1 | 8,885 | 255 |
| 44055 | analcatdata_supreme | 4,052 | 7 |
| 44056 | visualizing_soil | 8,641 | 4 |
| 45045 | delays_zurich_transport | 5,465,575 | 12 |
| 44059 | diamonds | 53,940 | 9 |
| 45046 | Allstate_Claims_Severity | 188,318 | 124 |
| 44061 | Mercedes_Benz_Greener_Manufacturing | 4,209 | 359 |
| 44062 | Brazilian_houses | 10,692 | 11 |
| 44063 | Bike_Sharing_Demand | 17,379 | 11 |
| 45047 | Airlines_DepDelay_1M | 1,000,000 | 5 |
| 44065 | nyc-taxi-green-dec-2016 | 581,835 | 16 |
| 45042 | abalone | 4,177 | 8 |
| 44066 | house_sales | 21,613 | 17 |
| 45043 | seattlecrime6 | 52,031 | 4 |
| 45048 | medical_charges | 163,065 | 5 |
| 44068 | particulate-matter-ukair-2017 | 394,299 | 6 |
| 44069 | SGEMM_GPU_kernel_performance | 241,600 | 9 |

Table 13: Categorical classification

| OpenML ID | Dataset Name | #Samples | #Features |
|---|---|---|---|
| 44156 | electricity | 38,474 | 8 |
| 44157 | eye_movements | 7,608 | 23 |
| 44159 | covertype | 423,680 | 54 |
| 45035 | albert | 58,252 | 31 |
| 45039 | compas-two-years | 4,966 | 11 |
| 45036 | default-of-credit-card-clients | 13,272 | 21 |
| 45038 | road-safety | 111,762 | 32 |

Table 14: Image Classification

| Dataset | #Train Samples | # Test Samples | #Categories | Input Size |
|---|---|---|---|---|
| MNIST | 60,000 | 10,000 | 10 | $28 \times 28$ |
| Fashion-MNIST | 60,000 | 10,000 | 10 | $28 \times 28$ |
| K-MNIST | 60,000 | 10,000 | 10 | $28 \times 28$ |
| Q-MNIST | 60,000 | 60,000 | 10 | $28 \times 28$ |
| CIFAR-10 | 50,000 | 10,000 | 10 | $32 \times 32$ |
| CIFAR-100 | 50,000 | 10,000 | 10 | $32 \times 32$ |

