# OpenReview forum: "On the Hyperparameter Loss Landscapes of Machine Learning Algorithms"
_ICLR.cc/2024/Conference — Submitted to ICLR 2024_

### Official Review · Reviewer_NkWq · 2023-10-16

**Soundness:** 3 good
**Presentation:** 3 good
**Contribution:** 3 good
**Rating:** 5
**Confidence:** 5

**Summary:**

Hyperparameter Optimization (HPO) is a well-established approach to tuning ML algorithms on a given dataset. Although there has been more than a decade of research on HPO, the underlying optimization problem is still poorly understood, and only a few papers study the HPO landscape and its properties. The paper at hand sheds further light on HPO characteristics by applying fitness landscape analysis (FLA) and a visual inspection approach to many HPO problems. Besides providing a more thorough analysis than previous papers, the authors are the first to provide a better understanding of fidelity landscapes and how the landscape changes with respect to the dataset. Overall, the authors confirm that HPO landscapes are fairly benign (in particular in a local neighborhood around the optimum), are often stable with respect to fidelities (e.g., sub-sampled datasets and less epochs), and are also consistent across datasets.

**Strengths:**

The paper is a very valuable contribution to a better understanding of HPO. So far, many HPO approaches are developed based on rough intuitions, sometimes by visualization of 2 hyperparameter interactions, and thus a sound foundation for HPO is missing. The paper contributes to further closing this gap and will allow the development of even more tailored HPO approaches.

The authors propose an entire workflow on how to study HPO landscapes in a systematic and thorough manner. It encompasses the interpretation of the search space as a graph, the dimensionality reduction to allow for a 2 dim-visualization, the use of fitness landscape analysis (FLA) and landscape similarities between fidelities and datasets.

A very important point is, furthermore, the thorough HPO data collection on 5 ML algorithms, each with more than ten thousand configurations and three / nine fidelity steps. This thorough data collection will allow further exploration and benchmarking of HPO.

The paper is overall very clear and easy to understand. (However, many typos are an issue – a spell checker is strongly recommended; also for the figures.)

**Weaknesses:**

Although I like the main direction of the paper a lot, there are many problems in the details – partially I raised them as part of the questions below.

1. Related work can be further improved. First, Xu et al. 2023 were not the first to treat HPO as a black box problem. I would be okay if you cite Bergstra and Bengio 2012 for that (although they might also not be the first ones). Second, Biedenkapp et al. [1] also proposed a 2-dim visualization of configuration spaces; a discussion on how the authors’ new approach refers to that is missing. Third, I don’t understand why Moosbauer et al. 2022 is cited for the nature of HP spaces. If the authors are unsure about what to cite for what, I recommend reading more surveys in detail.
1. The workflow is inefficient in practice because it assumes that the entire (discretized) configuration was evaluated. It would be much more relevant if it could be directly applied to the output of any HPO approach.
1. In view of the renowned paper by Bergstra and Bengio on random search and the low effective dimensionality of HPO spaces, I don’t understand why the authors decided to use a grid sampling strategy nevertheless and thus do not efficiently sample all dimensions. The distance function could be a reason for that, but defining a distance function in a non-discretized space is not uncommon.
1. The benchmarks heavily focus on tree-based algorithms. CNN is more of an outlier and the main paper discusses it also only very briefly (if at all). I would have hoped for a more diverse set of ML algorithms.
1. HPO benchmark libraries such as HPOBench or YAHPO already provide ample data (also for multi-fidelity). I wonder why the authors decided to collect new HPO data and do not validate their findings on these established benchmarks.
1. I wonder why the authors decided to use accuracy instead of balanced accuracy. I haven’t checked all the datasets from this paper, but unbalanced datasets are fairly common and thus balanced accuracy (or similar metrics) would be the better choice.
1. I’m not convinced that the no free lunch theorem plays a role here since, as the authors argue themselves, the characteristics of HPO landscapes seem to be very similar s.t. in fact a single algorithm could be the best one.
1. I’m not convinced by the general multi-fidelity conclusion in the way the authors state it. In fact, it is easy to construct a benchmark based on DNNs with a constant learning rate as the only hyperparameter and show that multi-fidelity HPO can fail in this case (if not done very carefully). Overall, the wording of the conclusions needs to be much more careful.
1. The source code and the data are not part of the submission. The authors write in their paper that they will release the data. However, from my experience, it is of utmost importance to provide all of this for the reviews in an anonymized way.

[1] Andre Biedenkapp, Joshua Marben, Marius Lindauer, Frank Hutter:
CAVE: Configuration Assessment, Visualization and Evaluation. LION 2018: 115-130

**Questions:**

* Portfolio approaches with complementary configurations (e.g., auto-sklearn 2.0 by Feurer at al.) are very famous and are constructed such that the configurations in the portfolio perform well on different datasets. If your conclusion is now that the area of well-performing configurations is consistent across datasets, I would also conclude that portfolio approaches should not be beneficial at all. One problem could be that the considered ML algorithms have this property, which might not be true for others; if that were true, it would partially invalidate your results. So, please comment on this contradiction between your results and previous results.
* What is a unit of the color coding in Figure 1? If these are accuracy benchmarks, the values cannot go up to 6000 and beyond.
* Why is the search space encoded as a ***directed*** graph? The neighborhood relationship goes in both directions.
* UMAP has several assumptions on the underlying space. Do all these assumptions hold true in this case?
* As you wrote, low effective dimensionality could have an impact on your evaluation. How would your results change if you first remove all unimportant hyperparameters?
* At the AutoML conference 2023, there was a paper on studying HPO landscapes for AutoRL by Mohan et al. How does your paper relate to that?

---

> ### Author Response · Authors · 2023-11-22
> **Response to Reviewer NkWq (Part 1)**
>
> We thank the reviewer for the insightful and useful feedbacks, please see the following for our response. Our revised version of the manuscript will also be updated soon.
>
> **(W1): Related work can be further improved.**
>
> We thank the reviewer to raise this question, we have proofread the references in our manuscript and correct inappropriate ones. In particular:
> - We cite Bergstra and Bengio 2012 for the formulation of HPO as a black-box optimization problem (**Section 1, page 1**, however, we note that in the submitted manuscript, we cite Xu *et al.* 2023 just for an example, and not to claim that they are the first one to do so.
> - We amended discussions about [1, 2, 3] on **Section 1, page 2**, as they represent the line of work that apply dimensionality reduction techniques to enable landscape visualization.
> - We removed the reference to Moosbauer et al. 2022 on HP search space.
> - We also corrected references at several other places.
>
> **(W2): The workflow is inefficient since it requires exhaustive search on a grid space. It would be desirable if it could be directly applied to the output of any HPO approach.**
>
> We appreciate the reviewer’s concern on this. However, we do need state that the goal of our developed FLA workflow is NOT to be competitive in efficiency as many other tools that work collaboratively with HPO methods (e.g., DeepCAVE, Optuna). Rather, our main motivation is to gain deep and comprehensive insights into the problem landscapes, which does not necessarily to be time-constrained. These two lines of work are often parallel to each other in landscape analysis literature. Therefore, we do not consider this should be a weakness of our proposed method.
>
> However, we do agree with the reviewer that it would be desirable by the community if our tool is able to serve as a post-optimization explanation method for existing HPO frameworks. We think this would be a promising direction to further improve it and thus make the advancements of landscape analysis accessible to a wider range of stakeholders.
>
> **(W3): I don’t understand why the authors decided to use a grid sampling strategy, which can be inefficient.**
>
> While we appreciate that grid search is less efficient from the HPO perspective, we contend that it is more favorable to landscape analysis. To be specific, the “low efficiency” of grid search as pointed out in the random search paper [4] mainly criticized its redundant sampling on configurations with highly repeated HP values. This is indeed not desirable when we wish to cover a more diverse set of configurations as in the HPO, or when we want to build up accurate models for the predicting model loss.
>
> However, for landscape analysis, such redundancy can actually be exploited to enable higher interpretability. For example, it allows our constructed landscapes to carry important information regarding “*how would the performance change if I only slightly modify the value of 1 HP*” without needing to use techniques like partial dependence plot, which are built upon surrogate models.
>
> **(W4): The benchmarks heavily focus on tree-based algorithms, and I would have hoped for a more diverse set of ML algorithms**
>
> To address the reviewer's concern, we include the analysis of HP loss landscapes of a feed-forward neural network (FCNet) on 4 UCI datasets, using part of the HPOBench data (**Appendix D, revised manuscript**). We also include supplementary results on NAS-Bench-101 and 201 to demonstrate the potential impact of our developed FLA framework for analyzing a wider range of AutoML tasks (**Appendix C, revised manuscript**).
>
> [1] Andre Biedenkapp *et al.*, "CAVE: Configuration Assessment, Visualization and Evaluation". *LION'18*, 2018.
>
> [2] Krzysztof Michalak, "Low-dimensional euclidean embedding for visualization of search spaces in combinatorial optimization", *IEEE Trans. Evol. Comput.*, 2019
>
> [3] Mathew J. Walter, *et al.* "Visualizing population dynamics to examine algorithm performance", *IEEE Trans. Evol. Comput.*, 2022.
>
> [4] James Bergstra and Yoshua Bengio, “Random search for hyper-parameter optimization”, *J. Mach. Learn. Res.*, 2012.

---

> ### Author Response · Authors · 2023-11-22
> **Response to Reviewer NkWq (Part 2)**
>
> **(W5): I wonder why the authors decided to collect new HPO data and do not validate their findings on established HPO benchmarks.**
>
> While we appreciate the considerable efforts that have made in developing an extensive set of HPO benchmarks, as incorporated in HPOBench, we choose to collect new data since we want a more *diverse* and *representative* set of scenarios that would better demonstrate the generalizability of our findings. In particular:
>
> - HPOBench considers 20 tabular datasets to evaluate tree-based models like Random Forest and XGBoost. We contend that the tabular benchmark with 57 datasets as used in this paper can lead to more statistical meaningful results.
> - We also consider the basic Decision Tree, which is the based model of tree ensembles, and LightGBM, which has been widely used in real-world engineering due to its faster speed.
> - For neural networks, HPOBench includes various NAS benchmarks, which we consider a different line of work from HPO, and simple models like MLP, which is not representative for more complicated DNNs.  We thereby seek to collect new data on CNNs.
>
> **(W6): I wonder why the authors decided to use accuracy instead of balanced accuracy, as there can be unbalanced datasets.**
>
> We report results on accuracy score since it is still the most widely used metric for classification. However, we do consider another metric for the evaluation of classifiers, ROC-AUC, to cope with potential dataset imbalance. Additionally, we had also done inspections on the 57 tabular data that are studied in this paper, and these datasets are actually all perfectly balanced between classes. We understand that the reviewer raises this problem since we only metioned accuracy in our submitted manuscript, and we have corrected this in our manuscript (**Section 4, page 5**).
>
> **(W7, Q1): Concerns regarding portfolio approaches and the No-Free-Lunch theorem.**
>
> We do not think there is contradictions here. While the main conclusion of this paper is indeed that the general family of HP loss landscapes of ML models share various high-level common patterns, we do note that the detailed landscape chracteristics still largely depend on specific model, search space, dataset, fidelity, etc. Therefore, in general, we can expect many methods to work relatively robustly across different tasks due to their high-level commonalities. However, there are still many differences between tasks that one may pay particular attention to, and thus make more informed selections on problem solvers. Portfolio approaches can leverage such information and lead to prominent results, while by no means should be considered useless according to our results.
>
> Based on this, we argue that the No-Free-Lunch theorem does have a role to play in HPO. However, concerning the fact that this paper focuses more on highlighting the similarities rather than nuances between HP loss landscapes, we remove the related expressoin from **Section 1 (page 1)** of the manuscript.
>
> **(W8): The authors should be more careful in their conclusions regarding multi-fidelity methods.**
>
> We thank the reviewer to raise this problem, and we agree that we should be more careful in making conclusions. In particular, we do note exceptions for our general findings. For example, there can be landscapes with hundreds of local optima, as shown by the outliers in **Figure 3**. Also, the long tail of the similarity distributions in **Figure 4** would imply there are cases when landscapes of the same model across fidelities and datasets can be very different. Nevertheless, these do not contradict with our main conclusions that HP loss landscapes tend to be benign, which are supported by the majority of the studied cases.  We have updated **Section 4** and **Section 5** of the manuscript to discuss this.
>
> As for the multi-fidelity conclusion that the reviewer specificly mentioned, as we discussed above, there can indeed have exceptions of our findings. However, based on our results on diverse models and datasets, we consider our conclusions that landscapes under different fidelities greatly resemble each other, generally hold for common cases. Our additional results on FCNets (**Appendix D**) and NAS-Bench-101/201(**Appendix C**) also supported this. To further demonstrate this, we plot the similarity between low-fidelitiy landscape with the full-fidelity one as a function of training epochs for the four FCNet scenarios (see this [anonymous dropbox link](https://www.dropbox.com/scl/fi/33u43dkg9p3koqy5l4x6v/iclr_fcnet_fidelity.pdf?rlkey=espeej682p7d86c9ewi2c31na&dl=0)).
>
> **(W9): Source code availability**
>
> Our group is highly committed to the reproducible research cultural. We will release the source code along with the dataset used and generated in this paper after the acceptance of this paper. To mitigate the reviewer’s concern on the generalization ability of our findings, we have provided additional experiments on NAS-Bench-101 and 201 (**Appendix C**).

---

> > ### Author Response · Authors · 2023-11-22
> > **Response to Reviewer NkWq (Part 3)**
> >
> > **(Q2): What is a unit of the color coding in Figure 1?**
> >
> > The unit in all landscape visualizations are the ranks of the performance. Therefore, for the CNN result, the range will be $[0, 6479]$ as its search grid contains $6480$ configurations. We report this instead of raw metric values because model losses/accuracies can be on very different scales under different scenarios (e.g., across datasets). We thank the reviewer for raising this problem, and we have updated all the related figures with explicit colorbars & captions to indicate this. We additionally note that for ease of interpretability, for both minimization (e.g., loss) and maximization (e.g., accuracy) cases, we calculate the ranks in the way such that lower rank values (e.g., 1st, 2nd, 3rd...) are better.
> >
> > **(Q3): Why is the search space encoded as a directed graph? The neighborhood relationship goes in both directions.**
> >
> > The edge directions and weights allows our constructed graph to model the propagation of $\Delta \mathcal{L}$ in the landscape, which can be exploited by graph mining techniques. For instance, PageRank centrality for each nodes typically show strong correlation ($0.5$ to $0.9$ for most cases) with configuration performance, despite it is calculated purely based on connectivity patterns of the network. Although we did not dive deep into this in the current paper, this already implies that the directed graph model of fitness landscapes can capture certain dynamic features of the landscape, and could be further exploited in the future.
> >
> > In addition, using this model, code-level implementation of landscape analysis methods can become more straight-forward. For example, since the edges in the graph always indicate the direction of improvement, first-improvement local search starting from a given node can be easily implemented by continuously following a random outgoing edge until a *sink* node (i.e., node with zero out-degree, local optimum) is encountered. Best-improvement local search, on the other hand, is almost the same, except that we need to select the outgoing edge with the largest improvement at each move.
> >
> > **(Q4): UMAP has several assumptions on the underlying space. Do all these assumptions hold true in this case?**
> >
> > There are three most important assumptions for UMAP, and are expected to hold in our case, at least with a good approximation. Here we discuss them in more detail:
> >
> > - *Data are uniformly distributed on Riemannian manifold*. As the distribution of HOPE embeddings essentially depends on the connectivity pattern of the graph, for HP loss landscapes, we can expect this assumption to hold. This is because the number of neighbors for each node in the graph would largely stay the same based on our neighborhood definition. Therefore, there will be no significance variation of density across the graph, and the HOPE embeddings in turn should approximately have a uniform distribution.
> > - *The Riemannian metric is locally constant.* The distances between HOPE embeddings correlate directly with the local connectivity patterns within the network. Given the stability in neighborhood structures as we discussed above, it is reasonable to presume that the local distance metrics would remain approximately constant in small regions.
> > - *The manifold is locally connected.* HOPE embeddings can actually preserve more than just the local structure between configurations, as it is able to capture high-order proximities in the graph. We thereby expect this assumption will also hold.
> >
> > Note that since HOPE is able to preserve asymmetric transitivity patterns in directed graphs, the first two assumptions may not be fully satisfied. However, considering the fact that the variations of in- and out-degrees of each node is limited by its total degree, which is almost identical across the landscape, and the fact that UMAP is designed to be able to operate robustly even when these assumptions only partially hold true, we think the application of UMAP here does not carry much risk.
> >
> > We have updated our manuscript with these discussions on UMAP in **Appendix B**.

---

> ### Author Response · Authors · 2023-11-22
> **Response to Reviewer NkWq (Part 4)**
>
> **(Q5): How would your results change if you first remove all unimportant hyperparameters?**
>
> As we metioned in the paper (**Section 4 on page 7**), we have already been selective in the HPs to study according to their importance, and we introduce our methodology in more detail in **Appendix F of the revised manuscript**. To be brief, all the HPs considered in our study have certain impact on model performance and complexity, and are among the commonly-used ones in literature and practical scenarios. We contend that incorporating only $2$ to $3$ most ones can lead to landscapes that fail to capture long-range HP interactions. Therefore, our studied scenarios typically involve $5$ to $8$ HPs for each model.
>
> Despite these, if we were to consider less HPs (leaving others to deafult values) with higher importance:
> - Neutrality of the landscape will decrease, as the remaining HPs are more likely to cause large performance shifts.
> - The number of local optima (if any) can also reduce due to the decrease in complex HP interactions (e.g., for XGBoost on dataset #45035, the number of local optima dropped from $64$ to $4$).
> - For NDC metrics, we do not observe statistically meaningful changes across our studied scenarios. The lower-dimensional landscapes still tend to be planar around the optimum.
> - However, for assortativity and autocorrelation, we observe a decrease in them for various landscapes when only studying the learning rate and maximum depth for XGBoost (e.g., for XGBoost on #45035, $ass: 0.7872 \to 0.7247, \rho_a: 0.6292 \to 0.4862 $). We postulate that one of the reasons for this is the fewer neutral regions in the landscape, which can form smooth plateaus and thus make the landscape more structured.
>
> **(Q6): At the AutoML conference 2023, there was a paper on studying HPO landscapes for AutoRL by Mohan et al. How does your paper relate to that?**
>
> This paper [1] investigates the RL hyperparameter landscapes, which is very different from the ML scenarios address in our work due to the dynamic nature of RL. In addition, it applies surrogate models based on RBF and GP to construct surrogate models of the fitness landscape, which has been a promising direction for studying expensive optimization problems (e.g., [2,3]).  This paper itself only considers aspects like the modality of the landscape, but as there have been increasing attention for studying HPs in RL (e.g., another paper in *ICML'23*, [4]), we believe it is a meaningful start to call for more in-depth landscape analysis on RL systems in future works.
>
> [1] Aditya Mohan *et al.*, “AutoRL Hyperparameter Landscapes”, *AutoML’23*, 2023.
>
> [2] Philip A. Romero *et al.*, “Navigating the protein fitness landscape with Gaussian processes”, *Proc. Natl. Acad. Sci. (PNAS)*, 2013.
>
> [3] Caroline J. Watson *et al.*, “The evolutionary dynamics and fitness landscape of clonal hematopoiesis”, *Nature*, 2020.
>
> [4] Theresa Eimer *et al.*, “Hyperparameters in Reinforcement Learning and How To Tune Them”, *ICML’23*, 2023.

---

### Official Review · Reviewer_1UHx · 2023-10-21

**Soundness:** 3 good
**Presentation:** 2 fair
**Contribution:** 1 poor
**Rating:** 5
**Confidence:** 4

**Summary:**

This paper performs a careful study on the HPs loss landscapes. It relies on modeling the hyperparameter configurations as a graph where every configuration is a node and an edge represents a similarity between two configurations. The authors conducted experiment on 5 search spaces and several datasets. Moreover, they compute several metrics such as autocorrelation, neutrality distance correlation, mean neutrality, no. local optima and mean basin size.

**Strengths:**

- The topic addressed by this paper is extremely important and interesting, as HPO is a common task for practitioners and scientists. The insights gained in this work help to understand better the HP spaces and design better HP optimizers.
- The method for evaluating the landscapes is interesting and uses helpful metrics.
- The conclusions and findings are coherent with previous work assumptions, so it is unlikely that a procedural error happened in this research.

**Weaknesses:**

- The study focuses on simple models such as random forest and support vector machines, while there is only a neural network architecture type involved (CNN). Given the relevance of neural networks in the current world, this study should include more detailed comparisons on this matter.
- Another common scenario is to create a broad space where both RF and SVM (algorithm and hyperparameter selection) are included. It is a missed opportunity to discuss how the loss structure is affected when optimizing both HP and algorithms.
- There should be a stronger discussion on the impact of the findings. From my perspective, although the findings are interesting, they are not surprising. Aspects like the correlation of the loss among fidelities and datasets are well-known and they are exploited by common HPO methods. In other words, it is not clear how the findings in this work can improve the current methods for optimizing HPs.

**Questions:**

There are some interesting points that I would be very interested in understanding:
1. How does the loss landscape change with the dataset size? Is there any available insight from this?

2. How does the loss landscape look when I consider only HPs that affect the expressiveness/complexity of the model? My intuition is that these HPs built up a convex search space, with not local optima. Is this somehow supported in this study?

---

> ### Author Response · Authors · 2023-11-21
> **Response to Reviewer IUHx (Part 1)**
>
> We thank the reviewer for the insightful and useful feedbacks, please see the following for our response. Our revised version of the manuscript will also be updated soon.
>
> **(W1): The paper focuses on simple models like RF and SVM, while more results on neural networks are desired.**
>
> We see there might be some misundertanding here, since our paper does not consider SVM. The submitted manuscript studies 5 ML models, including *i)*: Random Forest (RF), *ii)*: XGBoost, *iii)*: LightGBM, *iv)*:Deicision Tree, and *v)*: CNN.
>
> However, we agree with the reviwer that the community may be more desired to see results on neural networks. To this end, we have updated our manuscript with additional experiments on analyzing a feed-forward neural network setup from HPOBench using $4$ UCI datasets in **Section 4**. In addition, to demonstrate the potential of our proposed framework for analyzing more complicated DNN landscapes, we also conduct analysis on NAS-Bench-101 and 201 (**Appendix C**).
>
> **(W2): It is a missed opportunity to discuss how the loss structure is affected when optimizing both HP and algorithms.**
>
> Our proposed FLA framework can be easily adopted to include AutoML pipelines like model selection and pre-processing, just as it is able to analyze NAS landscapes by adjusting the neighborhood definition. However, as the main focus of this paper is on HPO, which is an well-established subfield of AutoML, such analyses is beyond its scope, and we leave this to future works.
>
> **(W3): There should be a stronger discussion on the impact of the findings, as studied aspects like multi-fidelity and transfer learning have been widely used in the community.**
>
> Despite transfer learning and multi-fidelity methods have been extensively used for HPO, they are mainly developed based on rough intuitions and their effectiveness are usually supported by algorithmic performance. To date, there is a lack of fundamental understanding regarding the validity of the underpinned assumptions made by these methods (see **Section 1, page 3, revised manuscript**). This has also pointed out by **Reviewer NkWq** in *Strengths*.
>
> This work fills this gap by providing a *problem-level* illustration on the success these methods are effective. In particular, we show that:
> - Multi-fidelity methods are reliable because the HP loss landscapes with lower fidelities greatly resemble high-fidelity landscapes w.r.t. both topological structure and performance ranks.
> - Transfer learning methods are effective since HP loss landscapes induced on diverse datasets also share considerable commonalities.
> - These findings are highly generalizable to a wide spectrum of tasks.
>
> Also, for the first time, we revealed an unified portrait of HP loss landscapes of ML models, which is quite benign. Such properties can be leveraged by more tailored algorithms to more efficiently traverse the landscape.
>
> In addition, the FLA framework we developed to enable all the analyses in our study, can also be applied to advance the understanding of a wider range of AutoML tasks, e.g., NAS (see analyses on NAS-Bench-101 and 201 in **Appendix C, revised manuscript)**.

---

> ### Author Response · Authors · 2023-11-21
> **Response to Reviewer IUHx (Part 2)**
>
> **(Q1): How does the loss landscape change with the dataset size?**
>
> We thank the reviewer to raise this question as we also find this is an interesting point that worth further exploration. Through amended experiments on the $57$ tabular datasets (**Appendix E, revised manuscript**), we find that with a weak correlation:
>
> - For all four tree-based models, the loss landscapes tend to be more similar on dataset with similar sizes.
> - In general, all $4$ tree-based models are more likely to overfit on datasets with more instances and features.
>
> However, here, we only applied most basic datasets meta-features like the number of instances and columns, while there are many more to explore (e.g., [1]). We think this should be investigated in more depth in future work.
>
> [1] Matthias Feurer *et al.*, “Initializing Bayesian Hyperparameter Optimization via Meta-Learning”, *AAAI’15*, 2015.
>
> **(Q2): How does the loss landscape look when I consider only HPs that affect the expressiveness/complexity of the model?**
>
> As we metioned in the paper (**Section 4 on page 7**), we have already been selective in the HPs to study according to their importance, and we introduce our methodology in more detail in **Appendix F of the revised manuscript**. To be brief, all the HPs considered in our study have certain impact on model performance and complexity, and are among the commonly-used ones in literature and practical scenarios. We contend that incorporating only 2 to 3 most ones can lead to landscapes that fail to capture long-range HP interactions. Therefore, our studied scenarios typically involve 5 to 8 HPs for each model.
>
> Despite these, if we were to consider less HPs (leaving others to deafult values) with higher importance:
> - Neutrality of the landscape will decrease, as the remaining HPs are more likely to cause large performance shifts.
> - The number of local optima (if any) can also reduce due to the decrease in complex HP interactions (e.g., for XGBoost on dataset #45035, the number of local optima dropped from 64 to 4). However, with the pressence of 4 local optima, this still somehow deviates from the reviewer's hypothesis that with less HPs, the landscape can be unimodal. We note that this is because the pressence of HP interaction (also known as epistasis in landscape literature). This leads to the fact that even if two HPs reveal exactly unimodal response, their combined effect can be multimodal due to their interactions. Nevertheless, as we see from the paper, most HP loss landscapes seem quite benign in terms of modality.
> - For NDC metrics, we do not observe statistically meaningful changes across our studied scenarios. The lower-dimensional landscapes still tend to be planar around the optimum.
> - However, for assortativity and autocorrelation, we observe a decrease in them for various landscapes when only studying the learning rate and maximum depth for XGBoost (e.g., for XGBoost on #45035, $ass: 0.7872 \to 0.7247, \rho_a: 0.6292 \to 0.4862 $). We postulate that one of the reasons for this is the fewer neutral regions in the landscape, which can form smooth plateaus and thus make the landscape more structured.

---

> > ### Comment · Reviewer_1UHx · 2023-11-23
> > **Reply to authors**
> >
> > I thank the authors for the response. I acknowledge that I have read it but still think that the contribution is very limited as the insights for the paper are not new for the community. Therefore, I decide to keep my score.

---

> > > ### Author Response · Authors · 2023-11-23
> > > **Reply to Reviewer IUHx**
> > >
> > > We respective the reviewer's opinion. However, we still would like to reiterate the contribution of this paper.
> > >
> > > Existing HPO methods often comprises 3 keys components: $i)$: a search space, $ii)$: an optimization strategy, and $iii)$: model evaluation. While $ii)$ and $iii)$ has received considerable attention in recent years, **the HPO search space, a.k.a. problem landscape, which plays a fundamental role in governing algorithmic behaviors, remain notably underexplored in our community.**
> > >
> > > To the best of our knowlede, we are the the first to provide  a **landscape-level** evidence to support the effectiveness of multi-fidelity and transfer learning methods, which though successful, are so far mainly **based on intuition**. It is previously **unclear why** and **to which extend** such methods are useful. We also analyze other landscape properties and discuss how they could be related to the success of existing methods. **Providing another view that could facilitate the transparency and interpretability of existing methods should be considered as a valuable contribution to the community.**
> > >
> > > More importantly, we believe with our developed FLA framework, this work has the potential to raise more attention on the application of landscape analysis to facilitate the understanding of a wider range of AutoML problems.

---

### Official Review · Reviewer_q8yg · 2023-10-27

**Soundness:** 2 fair
**Presentation:** 3 good
**Contribution:** 3 good
**Rating:** 5
**Confidence:** 4

**Summary:**

The authors propose a hyperparameter optimization (HPO) loss landscape analysis pipeline. This pipeline encompasses visualization techniques, landscape characteristics like neutrality and smoothness, and introduced landscape similarity metrics.
With this pipeline they analyze five classical machine learning (ML) models across different datasets and fidelities. In particular they connect landscapes resulting from the train loss to the test loss.
Their main findings are that the hyperparameter (HP) loss landscapes are fairly smooth and  unimodal, training and test losses of an HP configuration are mostly consistent except for configurations with a low training loss and that low-fidelity landscapes are fairly similar to high-fidelity landscapes propertywise.

**Strengths:**

### Originality
This paper is a significant leap in terms of hyperparameter loss landscapes analyses. While there are other existing papers and ideas (which the authors rightfully cite), this work represents a much broader, more detailed and nuanced study than the existing ones. Although they largely support the claims made in previous papers, having a broader study with a more extensive analysis pipeline certainly is an original contribution.

### Quality
The quality of this work is mixed. While I appreciate the great description of the experimental results and that the authors provide an extensive appendix detailing most of the studies performed, the paper has some questionable parts (see weaknesses).

### Clarity
The paper is generally structured in a clear and concise manner. Figures are easy to understand, but some important details are missing.

**Weaknesses:**

The work suffers from a set of problems regarding methodological choices in my opinion.

In particular, I see several weaknesses / questionable choices in the analysis pipeline:

1) Page 4 / HPO Loss Landscape Construction: The assumption that the search space is a grid and thus countable / finite is quite a strong one that should at least be discussed by the authors, imo. Real-valued hyperparameters such as learning rate are simply represented by a set of values potentially hiding many local optima that are not found often in the analysis of the authors. Closely connected to this weakness is the problem I see in the distance function $d$ defined by the authors. Assigning a distance on the basis of the distance of two values in the search space grid instead of their true numerical distance bears the danger of completely losing track of their true numerical meaning. Moreover, if I were to rearrange the values in the grid such that they are no longer numerically ordered, the resulting neighborhood and thus graph would change - something that seems very odd to me. Similarly, I find it odd that the corresponding graph does not have improving edges between configurations differing only in one hyperparameter if their numerical values are not right next to each other in the search space grid. One may wonder whether it is sensible to have an edge between a configuration with a learning rate of 0.1 and 0.01 but not between 0.1 and 0.001, if both lead to improvements. The authors frame this encoding as a strength, which I disagree with. In particular, it is unclear to me to what degree this influences the final results of their findings as it might have an inherent smoothing effect caused by prior knowledge encoded via the search space. Most HPO tools do NOT follow such a discretization strategy such that I am not sure whether all of the insights gained here are really helpful for designing better HPO tools. Similarly, it is unclear what influence the hyperparameter of the framework defining when a neighbor is a neutral neighbor has on the results. What if this values was set to $2\%$ or $5\%$? Would the results look differently?

2) Page 4 / Landscape Visualization: In the last sentence, the authors state that they apply a linear interpolation to generate a smooth surface. At the same time, they argue at the end of page 5 that the loss landscapes shown in Figure 3 (a) are “relatively smooth”. It is unclear whether this is an effect of the smoothing applied or would be the case anyway. Considering that the FLA metrics concerning smoothness show similar results, it will probably also apply without the smoothing step. Nevertheless, this seems a little questionable to me.

3) Page 7/ Highly neutral; planar around the optimum: The authors state that they have been “selective in the search space design”, which is not further explained. They do list the search spaces in the appendix, but considering this comment, it seems that the authors took great care in choosing these search spaces which naturally greatly influences the results presented in the paper. Thus, the authors should comment on their approach for choosing the search space and on the consequences such as potential limitations of their analysis. They should also relate this to the setups of related analyses they cite.

4) Limitations: In general I miss an elaboration on the limitations of the authors’ analysis. For example
    * Can conditional dependencies between hyperparameters be modeled in the framework?
    * Do the embeddings account for the fact that for numerical hyperparameters (in principle) values between those given in the search grid can be chosen, but for categorical hyperparameters not? This is important as the analysis otherwise inherently assumes all hyperparameters to be categorical, (at least to a large extent).
    * What hyperparameters for UMAP were set? This greatly influences the outcome of the analysis (see e.g. https://pair-code.github.io/understanding-umap/), but the concrete hyperparameters or how they were chosen are never mentioned.

It is also very unfortunate that no source code is provided since this makes the claim of the authors in the conclusion that their framework could be used to analyze other benchmarks a bit meaningless.

Furthermore, the authors could highlight a little bit more that also other papers performed dimensionality reduction techniques to show how loss landscapes look like. I agree that the approach of using HOPE + UMAP for this has its pros and is novel, but easier approaches have been used before (see e.g. Q2 on page 5 in [A]).

Besides the problems explained above, the paper suffers from some (easy to fix) inconsistencies and typographical errors.
* Page 4 & 9: “tunning” -> tuning
* Page 5: “[...]under different scenarios, We apply[...] -> scenarios. We
* Page 15: “knoledge” -> knowledge
*Page 17: Paragraphs on HOPE and UMAP have quite some grammatical errors.
* The references are largely inconsistent. For example, the first names of the first reference are not abbreviated, the ones of the second one are. This applies to many more.ArXiv papers are sometimes references as “arXiv preprent arXiv:...” and sometimes as “CoRR, abs…”. The paper cited for “A survey of methods for automated algorithm configuration” by Schede et al. is only the extended abstract, although a full version is published in JAIR: https://www.jair.org/index.php/jair/article/view/13676/26852

[A] Sass, René, et al. "Deepcave: An interactive analysis tool for automated machine learning." arXiv preprint arXiv:2206.03493 (2022).


EDIT AFTER REBUTTAL: Slightly increased score.

**Questions:**

Since most of my questions are related to the problems I see in the methodological setup, I have listed them under weaknesses. However, I have some more questions here that are not connected to a weakness:
* Can you comment about the consistency of landscapes across seeds?
* From Figure 6 we can see that the correlation between test and train performance is very good for configurations with a bad training performance, but less so for those ones with good training performance. Even more importantly, it does not seem to be the case that performing some kind of early stopping would counter an overtuning effect. Do you have concrete suggestions how HPO tools should be changed based on these findings?
* In the conclusion you mention that new HPO tools could be designed based on your findings. Do you have exemplary ideas? I wonder whether this is really the case since your findings largely are coherent with existing knowledge from smaller studies.

I am more than happy to adjust my rating, if the authors do the following:
* Clearly comment on the consequences of their design choices which I criticize under weaknesses in 1) - 3). In particular, I would at least like to know what limitations come with their design decisions, but ideally how that influences the results concretely.
* Add a section on the limitations of their analysis (see critique point 4).

Of course, I am also more than happy to increase my rating, if the authors or other reviewers point out factual mistakes in my review. To be very clear: I will raise my score to Accept, if I deem the problems 1) - 3) sufficiently addressed, as I think that the paper is solid except these problems.

---

> ### Author Response · Authors · 2023-11-22
> **Response to Reviewer q8yg (Part 1)**
>
> We sincerely appreciate the reviewer's recognition of the overall contribution of this work and we address the raised concerns as follows:
>
> **(W1): Concerns regarding the use of grided search space.**
>
> - *"Simply representing real-valued HPs by grid values can potentially hiding many local optima"*: Grid-based search spaces may not necessarily lead to the miss of local optima, and when it does so, we contend that the influence on our results is acceptable. To explain this idea, let's image a 1D space, where a local optimum lies between two grid points.
>   - *If the basin of attraction of this local optimum is larger than a grid unit:* (see [an illustration](https://www.dropbox.com/scl/fi/o38vuraconlxrvbyo5h9c/grid_lo_1.png?rlkey=2gafrhjhtshk8jiog6petqzax&dl=0)) Then, one of the nearby grid-points will be identified as the center of this basin. In this case, we end up with an approximation of this local optimum rather than miss it. The precision of such approximation depends on our grid size, and we note that for continuous landscapes, we can never truely pinpoint the local optimum.
>   - *If the basin of attraction of this local optimum lies in a grid unit:* (see [an illustration](https://www.dropbox.com/scl/fi/z41is1dfe83kuxigzcal4/grid_lo_2.png?rlkey=5p06085hdej6q822dif29cs19&dl=0)) In this case, we do bear the danger of missing it. However, considering its tiny basin, this would not cause much impact to our modality conclusion.
> - *"if I were to rearrange the values in the grid such that they are no longer numerically ordered, the resulting neighborhood and thus graph would change - something that seems very odd to me"*: We respectively oppose the reviewer’s comment at this point, since it would be odd to set up a grid space in which values do not follow their numerical orders. Please see [an illustration](https://www.dropbox.com/scl/fi/vq7ptu5up7c4gb5248hm0/grid_neighbor.png?rlkey=60acujx3kss34y9ih1947gn2e&dl=0) of our 2D grid space.
> - *"I find it odd that the corresponding graph does not have improving edges between configurations differing only in one hyperparameter if their numerical values are not right next to each other in the search space grid."*: Using the same 2D illustration as above, the neighborhood $\mathcal{N}$ of a configuration (e.g., $\lambda_{2,2}$ in the example) is the four configurations with 1-edit distance from it (as pointed by the arrows). This is consistent with the classic fitness landscapes in evolutionary biology, as well as as our intuitive understanding for 'neighborhood'. However, if we were to allow for improving moves from a learning rate of $1 \times 10^{-4}$ directly to $1 \times 10^{-1}$ (i.e., $\lambda_{2,2} \to \lambda_{5,2}$ in the example), this will form an odd neighborhood. Also, we think it is not common for human experts to make such big moves during manual HP tuning.
> - *"Most HPO tools do NOT follow such a discretization strategy such that I am not sure whether all of the insights gained here are really helpful for designing better HPO tools."*: While we agree with the reviewer’s opinion that grid spaces is not used by common HPO methods, we argue that we do not intend to use this grid-based landscape to simulate the *exact* algorithmic behavior. Instead, we only apply the collected grid points to obtain a sketch of the landscape, and thus provides high-level *guidance* to the searching process.
> - *"it is unclear what influence the hyperparameter of the framework defining when a neighbor is a neutral neighbor has on the results.":* The the threshold $\epsilon$ that we choose to define neutral neighbors is related to the neutrality and NDC metric.
>   - For landscape neutrality, we provide [a plot]((https://www.dropbox.com/scl/fi/80isdxig6s6qm5r7xiv72/iclr_fcnet_neutrality.pdf?rlkey=fu4r63hgaq0rrbvla5464u90c&dl=0)) of the cumulative distribution of the relative performance between neighboring configurations, which clearly demonstrates how the choice of $\epsilon$ can affect the result. As we can see, small improvements are quite prevalent in this landscape. We choose to report the results using $\epsilon=0.001$ only because differences below this level can make no practical impact for most scenarios.
>   - For the NDC metric, we also developed an alternative way for calculating it, which relies on the raw pairwise performance differences rather than node neutralities (**Appendix B.2**). Using this $\epsilon$-agnostic NDC metric, we obtain highly consistent results as we report in Section 4. That is, the improvements at each step on an adaptive walk towards the optimum would continuously decrease. We choose to report the neutrality-based NDC scores in this paper only to keep a consistency with the neutrality definition.
>   - Final remark: the neutrality in a coninuous landscape can vary with the way we define it. In contrast, the 'planar around optimum' conclusion is an intrinsic property of the landscapes, and is thus more robust to our metrics for measuring it.

---

> ### Author Response · Authors · 2023-11-22
> **Response to Reviewer q8yg (Part 2)**
>
> **(W2): It is unclear whether the smooth landscape as shown in visualization is an effect of linear interpolation.**
>
> We understand the reviewer’s concern. We provide an updated diagram showing the workflow of our proposed visualization method in **Figure 7 (Appendix B)**, in which a scatter plot of the data points before interpolation is provided. It is clear to see that the smoothness conclusion does not directly hinges to the linear interpolation. We do this step only to refine the interpretability of the plots.
>
> **(W3): The authors should comment on their approach for choosing the search space and on the consequences such as potential limitations of their analysis. They should also relate this to the setups of related analyses they cite.**
>
> We have updated our manuscript with details about the routines we take in search space design in **Appendix F**. In general, we design these search spaces to combine both knowledge from existing benchmarks/literature and empirical experience. However, in addition to HP importance, as we only have limited computational budget (though in total we spent tens of thousands of GPU/CPU hours), we cannot include all the HPs that have an observable impact on performance. On the other side, we also do not want our search spaces to be too simple, for example, a search space containing only 2 HPs can be more likely to reveal a unimodal structure due to less high-order HP interactions. Bearing these considerations in mind, our designed search spaces eventually contain 5 to 8 HPs for each model, spanning 6,480 to 24,200 configurations in total.
>
> In other related works, Pushak and Hoos [1] also considers XGBoost, where they evaluated a total of $413,343$ configurations for 11 HPs, but only on a single dataset. They also analyzed the same FCNet benchmark data as in our amended experiments (**Appendix D**). Rodrigues *et al*. [2] analyzed 5 HPs of CNN using merely $800$ configurations in total. Teixeira and Pappa [3], and also Pimenta et al. [4] studied AutoML search spaces, which are generated using derivation trees from a proposed context-free grammar, and are not directly comparable to HP search spaces. Schneider et al. [5] compared the landscapes of XGBoost and classic black-box optimization problems using XGBoost search spaces with $\{2,3,5\}$ HPs, using mostly logarithmic-valued grids.
>
>
> While we believe our designed spaces are representative of real-world scenarios, and are not manually biased towards certain conclusions, we do note that our FLA metrics can be affected by the selection of HPs and their grid spacing.
>
> For example, for the neutrality metric, using an extremely dense grid for HPs, e.g., $lr=[0.1, 0.10001, 0.10002, 0.10003, \dots]$ can lead to landscapes with extremely high neutrality,  whereas considering only the most important 1 or 2 HPs can shape landscapes with little neutrality. Landscape properties as measured by other metrics can also slightly vary with the search space design, and for detailed discussion on this, we kindly refer the reviewer to our response to **reviewer IUHx** (**Part 2, Q2**). Considering the extensive set of scenarios that we have studied, and the amended  experiments on existing benchmarks (HPOBench and NAS Bench-101/201, **Appendix C and D**), our result should be generalizable to the majority of practical cases.
>
> [1] Yasha Pushak and Holger Hoos, “AutoML loss landscapes”, *ACM Trans. Evol. Learn.*, 2022.
>
> [2] Nuno M. Rodrigues, “Fitness landscape analysis of convolutional neural network architectures for image classification”, *Info. Sci.*, 2022.
>
> [3] Matheus C. Teixeira and Gisele L. Pappa, “Understanding AutoML Search Spaces with Local Optima Networks”, *GECCO’22*, 2022.
>
> [4] Cristiano G. Pimenta *et al.*, “Fitness Landscape Analysis of Automated Machine Learning Search Spaces” , *EvoCOP’20*, 2020.
>
> [5] Lennart Schneider *et al.*, “HPO × ELA: Investigating Hyperparameter Optimization Landscapes by Means of Exploratory Landscape Analysis”, *PPSN’22*, 2022.

---

> ### Author Response · Authors · 2023-11-22
> **Response to Reviewer q8yg (Part 3)**
>
> **(W4): In general I miss an elaboration on the limitations of the authors’ analysis.**
>
> We thank the reviewer to raise this problem. We have updated the final section of the body of the paper to include some limitation of our study. Here we also response to the 3 points that the reviewer specifically mentioned:
>
> - *Can conditional dependencies between hyperparameters be modeled in the framework?* Yes. Conditional dependencies can be incorporated into landscapes by defining distances using grammar derivation tree (see, e.g., [1]). Actually, our framework is designed to be able to operate on the majority (if not all) of AutoML scenarios by properly defining a neighborhood structure. We show such an example on NAS-Bench-101 and 201 in the updated manuscript (**Appendix C**).
> - *Do the embeddings account for the fact that for numerical hyperparameters (in principle) values between those given in the search grid can be chosen, but for categorical hyperparameters not?*: Yes, HOPE generates embeddings of nodes based on the local connectivity of them. For numerical HPs, if intermediate values are added to the search space, the resulting new configurations would be closely connected to its neighbors that previous exist in the landscape. The generated embeddings would thus also stay between the existing configurations in the landscape in the embedding space, since HOPE is able to preserve local structure of the network.
> -  *What hyperparameters for UMAP were set?*: While the exact setting of UMAP should depend on the size of the landscape (e.g., total number of configurations), we have added a remark to provide some general guidelines for UMAP setup for landscape visualization purpose in **Appendix B**
>
> **(Additional Comment): Source code availability**
>
> Our group is highly committed to the reproducible research cultural. We will release the source code along with the dataset used and generated in this paper after the acceptance of this paper. To mitigate the reviewer’s concern on the generalization ability of our findings, we have provided additional experiments on NAS-Bench-101 and 201 (**Appendix C**).
>
> **(Additional Comment): The authors could highlight a little bit more that also other papers performed dimensionality reduction techniques.**
>
> We thank the reviwer for this advice. We have added comments on works that directly applies dimensionality reduction techniques in the **Page 2 of Section 1**. To be brief, a direct use of dimensionality reduction methods can cannot well-preserve local connectivity pattern in the landscape, and would thus lead to results that are less like a “topographic map”. In addition to the reference that the reviwer provides, we also mention other works in this direction, e.g., [1, 2].
>
> **(Q1): Can you comment about the consistency of landscapes across seeds?**
>
> We statistically find that the landscapes across seeds are highly consistent with other in terms of variance.
>
> **(Q2): Do you have concrete suggestions how HPO tools should be changed based on the findings in Figure 6?**
>
> We contend that actually an early stopping strategy can be effective to deal with those configurations with small training loss but poor generalization performance by using a validation set. To demonstrate this, here we show the [learning curve](https://www.dropbox.com/scl/fi/6zb62cd8qgmal6wifxh3q/iclr_fcnet_curve.pdf?rlkey=o5dsjp69m2m2mo85r4dd1k3k4&dl=0) (note that in **Figure 6**, each point represents the *final* performance of each configuration) for 3 representative points in **Figure 6**. Here, the *good* one is the best configuration. We can clearly see that the its validation loss gradually decreases as the training proceeds. After around 250 epochs, there is barely no further improvement (and thus the algorithm can terminates by using early stopping). For the other two configurations ('middle' and 'poor'), we see that the validation loss quickly rises with model training, and eventually stabilize at some level. Such learning curves reveal significant different patterns as compared to the prominent one. Therefore, early stopping technique that discards configurations with increasing test-loss during the early phase of training can be effective to address such problems.
>
> [1] Krzysztof Michalak, “Low-Dimensional Euclidean Embedding for Visualization of Search Spaces in Combinatorial Optimization”, *IEEE Trans. Evol. Comput.*, 2019.
>
> [2] Mathew J. Walter *et al.*, Visualizing population dynamics to examine algorithm performance. *IEEE Trans. Evol. Comput.*, 2022.

---

> ### Author Response · Authors · 2023-11-23
> **Response to Reviewer q8yg (Part 4)**
>
> **(Q3): In the conclusion you mention that new HPO tools could be designed based on your findings. Do you have exemplary ideas? The current findings largely are coherent with existing knowledge from smaller studies.**
>
> Since we find that the properties studied in this paper are highly prevalent for the majority scenarios, we contend that auxiliary techniques such as search space pruning, initialization with transfer learning and multi-fidelity, which are agnostic of the main searching strategy, can be effectively combined to form a framework that can facilitate the efficiency of general optimization methods. In addition, strategies that is able to better navigate in highly neutral regions, and with adaptive strides, may probably also lead to improvements.
>
> We agree with the reviewer that these techniques are not new to the community. However, previously, they are **mainly based on intuition and are studied in isolation**, and there is no concrete evidence to support the general feasibility of these methods from a fundamental (i.e., problem landscape) perspective. This paper for the first time provides solid empirical evidence to demonstrate that these techniques are widely applicapable **in a combined manner due to the intrinsic properties of their HP loss landscapes.**
>
> In addition, while in this paper we begin from investigating high-level properties of HP loss landscapes, we believe our proposed FLA framework, which will be continuously updated with new analytical methods, can serve as the cornerstone to facilitate the understanding of more detailed landscape topographies.

---

### Official Review · Reviewer_N7RA · 2023-10-29

**Soundness:** 2 fair
**Presentation:** 3 good
**Contribution:** 2 fair
**Rating:** 8
**Confidence:** 4

**Summary:**

This paper studies HP loss landscape across several important scenarios: train v.s. test; fidelity; datasets and models. To conduct the analysis, the authors invented a new dimension reduction techniques for HP analysis: it first defines a DAG where the nodes are HPs and edges are “improvement edges” so that an edge from v_i to is neighbor v_j (neighbor defined based on a distance function) means the loss of v_j is smaller than v_i. Then it applies the HOPE method to learn the node embedding and further uses UMAP to reduce the dimension to 2. Based on the resulting landscape, visual inspections and many metrics defined in the Fitness Landscape Analysis (FLA) literature can be used to characterize the landscape. Four observations, though many are not very surprising, are derived.

**Strengths:**

The paper is mostly clearly written and the empirical analysis seems quite solid. It also proposed a new dimension reduction method for such analysis based on graph neural networks, which to the best of my knowledge, the first work doing so. They also use the metrics from the FLA field to characterize the landscape, which is also novel to me.

**Weaknesses:**

This work provides empirical evidence that many HPO practitioners already use, which limits the work’s impact and significance. For example, the finding that HP landscapes are highly similar across datasets is the basis for hyperparameter transfer learning. The similarity across fidelity verifies again that the assumption of multi-fidelity HPO methods.

**Questions:**

What’s the motivation of the proposed method in Section 2? What’s the pros and cons compared to other related works? Why do the authors think the proposed method is better than the others such as Pushak & Hoos (2022)? Like mentioned in the section of “Nearly unimodal”, the conclusion may indeed different based on what analysis method to use. If using the method from Pushak & Hoos (2022) in the same setting as in this paper, would the authors have the same conclusion? This may be a good experiment to cross verify the findings.

I may have missed this. What is the space for quantifying the landscape characteristics? The low dimension embedding based on HOPE method, or the 2d after UMAP?

If many observations in Section 4.1 are based on the 2d landscape, then my following question will be relevant. In Figure 3, the landscape looks indeed smooth. But is it a result of using graph embeddings and UMAP, which smooth the landscape? Can it be different in the original higher dimensional space? Also can other properties of the landscape also due to the usage of graph embeddings and UMAP? I don’t know much about high dimensional statistics and if there are some methods to quantify characteristics in the original HP space, that would be great. Otherwise, successful application of the observations derived from the paper to some HPO application could also in-directly verify the findings are relevant. For example, if the optimal tends to have a plateau, then the local search step in BO should not bring much improvement.

Since the author mentioned “However, when zooming into the top-10% regions, we find that the majority of our studied scenarios reveal low γ-set similarities.”, I think this should be highlighted in the paper because the prominent regions are the areas that people care about.

---

> ### Author Response · Authors · 2023-11-21
> **Response to Reviewer N7RA (Part 1)**
>
> We thank the reviewer for the insightful and useful feedbacks, please see the following for our response. Our revised version of the manuscript will also be updated soon.
>
> **(W1) This work focuses on aspects that many HPO practitioners already use (e.g., multi-fidelity, transfer learning)**
>
> Despite transfer learning and multi-fidelity methods have been extensively used for HPO, they are mainly developed based on rough *intuitions* and their effectiveness are usually supported by algorithmic performance. To date, there is a lack of fundamental understanding regarding the validity of the underpinned assumptions made by these methods. This has also pointed out by **Reviewer NkWq** in Strengths.
>
> Our work fills this gap by providing a *problem-level* illustration on the reason that these methods are effective. In particular, we show that:
> - Multi-fidelity methods are reliable because the HP loss landscapes with lower fidelities greatly resemble high-fidelity landscapes w.r.t. both topological structure and performance ranks.
> - Transfer learning methods are effective since HP loss landscapes induced on diverse datasets also share considerable commonalities.
> - These findings are highly generalizable to a wide spectrum of tasks.
>
> Also, for the first time, we revealed an unified portrait of HP loss landscapes of ML models, which is quite benign. Such properties can be leveraged by more tailored algorithms to more efficiently traverse the landscape.
>
> In addition, the FLA framework we developed to enable all the analyses in our study, can also be applied to advance the understanding of a wider range of AutoML tasks, e.g., NAS (see analyses on NAS-Bench-101 and 201 in **Appendix C, revised manuscript)**.
>
> **(Q1) What is the motivation, as well as pros & cons of the proposed landscape analysis method?**
>
> From the highest level (see updated conceptual diagram in **Figure 7, revised manuscript**), our proposed methods are motivated by the fact that prior works only used an limited set of FLA metrics to assess partial HP loss landscape characteristics, and could not provide intuitive visualization of landscape topography, or to interrogate the similarities between different landscapes. These gaps are also mentioned in **Section 1 on page 2**. Consequently, they fail to provide an unified, comprehensive understanding of HPO search spaces.
>
> Our proposed method fills these gaps by (also see **Section 1 on page 2**):
> 1. Developing a novel landscape visualization method that can preserve both local and global landscape structure.
> 2. Synthesizing a diverse set of FLA metrics to characterize HP loss landscapes from several essential, complementary perspectives.
> 3. Applying 3 dedicated landscape similarity measures tailored for HPO context (by leveraging configurations ranks) to enable informative comparison between landscapes.
>
> Next, we address the reviewer's concern by providing a more detailed discussion on their pros and cons:
> 1. *Landscape visualization:* Our proposed visualizatoin method is able to handle high-dimensionality landscapes while incorporating neighborhood information, and thus allows us to provide a highly interpretable illustration of landscape topography that none of the priors works is able to. However, we do note that we need to be cautious when interpreting the result, as any 2D visualization of high-dimensional data can inevitably loss certain details. This paper ensures the correctness of the observed patterns using FLA metrics that are directly calculated on the original HP space.
> 2. *FLA metrics:* Our applied metrics together enables the first unified portrait of HP loss landscape topography. However, while we consider this feature set sufficient for the purpose of this work, it could be further enriched in the future with metrics characterizing more detailed properties of the landscape.
> 3. *Landscape similarity measure:* Our method enables the first comparison between different HP loss landscapes, and the results are directly concerned by HPO solvers. However, such methods require the landscapes under inspection to have exactly the same set of configurations. In future work, this could be addressed by using a portfolio of landscape features and graph-level embeddings, which can be calculated on landscapes on different scales.
>
> Finally, for the modality part that the reviewer had specifically mentioned, we note that this is rather a factual problem instead of method-dependent one, as any FLA methods should only faithfully reflect the true underlying structure of the landscape. Therefore, our assessment of landscape modality is in essence the same as in Pushak & Hoos (2022). In fact, for the amended experiments on FCNet (**Appendix E, revised manuscript**), we observe very similar results as in their. However, we also note that other aspects that we studied, e.g., basin of attraction, neutrality, smothness, are not considered in their work.

---

> ### Author Response · Authors · 2023-11-21
> **Response to Reviewer N7RA (Part 2)**
>
> **(Q2) What is the space for quantifying the landscape characteristics?**
>
> All the FLA metrics we calculated in this study directly operate on the *original* high-dimensional landscape. The HOPE embeddings and UMAP method are only used for generating landscape visualizations (see an updated conceptual workflow with more details in **Figure 7, revised manuscript)**.
>
> **(Q3) Is the smooth landscapes as can observed in the visualizations caused by the use of UMAP or HOPE?**
>
> We verified all the findings observed from landscape visualizations using a set of dedicated FLA metrics (**Table 1, page 4**). Importantly, these metrics are exactly developed to operate on the original high-dimensional space, and do not rely on embeddings in low-dimensional space. Therefore, a good match between qualitative and quantitative results as in this paper would imply that our landscape visualizations can faithfully preserve patterns in the high-dimensional landscape, and our findings are thus not a result of HOPE, UMAP or interpolation methods.
>
> For example, to verify that there is a flat plateau around the global optimum, we calculate NDC metrics for our studied landscapes. High NDC (**Figure 3 (c), page 6**) values would imply that as we move closer to the global optimum, we are more likely to encounter local search moves with subtle improvements. Beyond our study, experiments on HPO methods in many existing literature also have observed diminishing improvement near the best-found configuration (e.g., [1,2]).
>
> [1] Jasper Snoek *et al.*, “Practical Bayesian Optimization of Machine Learning Algorithms”, *NIPS’12*, 2012.
>
> [2] Stefan Falkner *et al.*, “BOHB: Robust and Efficient Hyperparameter Optimization at Scale”, *ICML’18*, 2018.
>
> **(Q4) The authors should highlight the results on the top-10% region of the landscape**
>
> We thank the reviewer for this helpful suggestion. We have updated the related expression in our revised manuscript.

---

> > ### Comment · Reviewer_N7RA · 2023-11-22
> > **Response to the authors**
> >
> > I would like to thank the authors for their detailed response. I missed that the FLA metrics are computed in the original space.
> >
> > In general I agree with the authors' response. The paper provides support for many assumptions we use in transfer learning and multi-fidelity methods. The analysis using FLA metrics are very useful in practice and the visualization technique applied to HPO search space is novel. I think the proposed methods and generated insights are worth to share in the HPO community. I believe this paper is a solid work on the landscape analysis and will increase my score.

---

> > > ### Author Response · Authors · 2023-11-23
> > > **Appreciate the reviewer's recognition**
> > >
> > > We sincerely appreciate the reviewer's recognition for this work, and thank you for your valuable time for providing the detailed comments and reading our response.

---

### Author Response · Authors · 2023-11-23
**Global Response**

We would like to thank all the reviewers for their constructive feedback and time in the reviewing process. We have provided detailed response individually to address every concern raised in the comments. We hope that all the raised concerns have been adequately addressed. We would greatly appreciate any additional feedback you may have, and we can discuss them openly.

---

### Author Response · Authors · 2023-11-23
**Revision History**

We would like to express our gratitude to each reviewer for their insightful and helpful feedback. We have provided corresponding responses to each reviewer. Now, we summarize the **major** updates made in the revised version of the paper to facilitate reviewers in finding the new content of interest:

- **Section 1, page 2:** For the first bullet point on the limitations of existing landscape visualization method, a discussion on methods that directly used dimensionality reduction techniques has been added.
- **Section 1, page 3:** For the paragraph on multi-fidelity HP loss landscapes, further emphasis on the lack of problem-level evidence to support their effectiveness has been added.
- **Section 3, page 5:** For the experimental setup, scenarios for a feed-forward neural network and NAS-Bench-101/202 have been added. All the related information in the abstract and other sections have also been updated.
- **Section 4:** Across section 4, comparisions of HP loss landscapes and NAS loss landscapes have been added wherever needed.
- **Section 4, page 6:** Added a discussion on the modality of the feed-forward neural network scenarios.
- **Section 4, page 7:** Added a discussion on the dependency of the neutrality conclusion on the choice of search space.
- **Section 4, page 8:** Added a note on the potential relationship between landscape characteristics and dataset meta-features.
- **Section 5, page 9:** The conclusion part has been updated with discussions on the exceptions of our findings and limitations of the proposed analysis method, as well possible directions for future works.
- **Appendix B, page 17:** Added an updated diagram showing the workflow of our proposed method.
- **Appendix B.1, page 17:** Added discussions on the validity of the three assumptions made by UMAP method in our use case.
- **Appendix B.1, page 18:** Added remarks on our reasoning for choosing HOPE and UMAP, as where as the hyperparameter setting of UMAP.
- **Appendix B.2, page 19:** Added an alternative method for calculating the NDC metric, which is independent of the definition for neutrality.
- **Appendix B.3, page 20:** Added an introduction of the local optima network (LON), which is applied in the analysis of NAS-Bench-101/201 and feed-forward neural networks.
- **Appendix C, page 21-25:** Added a new appendix section, providing extended analysis on NAS-Bench-101/201 loss landscapes using our proposed FLA framework.
- **Appendix D, page 25-26:** Added a new appendix section, providing extended analysis on HPOBench data (a feed-forward neural network) using our proposed FLA framework.
- **Appendix E, page 26-27:** Added a new appendix section, providing further explorations on the relationship between loss landscapes structures and dataset chracteristics.
- **Appendix F, page 27-28:** Added the routines we take in designing the search spaces used in this work.

---

### Meta-Review · Area_Chair_xjKd · 2023-12-10

**Metareview:**

This work studies hyperparameter loss landscapes of ML models. The study considers 60+ data sets , 5 ML models, and 11 million evaluations at different fidelities. The empirical results suggest landscapes a fairly smooth with large plateaus near the optimum. The results also suggest the datasets share considerable similarities that can be exploited to accelerate HPO, at least for the model considered in the study. Conducting a study of this type is timely and can provide insight of great value to the community.

Conducting such a study is also consuming in terms of time and effort, and I would like to praise the authors for raising to the challenge. Unfortunately, the majority of the reviewers raised concerns about the conclusions drawn from the reported results. In particular, the reviewers remained unconvinced after the rebuttal of the design choices made in this study. Another weakness of the study is that it focuses on simple models such as random forest and support vector machines. Given the relevance of neural networks today, this feels like extending the the study into the realm of deep neural network would have strengthen the work.

**Justification For Why Not Higher Score:**

While work of this type should be encouraged and praised, the design choices made by the authors were unfortunate. Also, the findings were not surprising and in line with work published elsewhere.

**Justification For Why Not Lower Score:**

N/A

---

### Decision · Program_Chairs · 2024-01-16

Reject